# MIPAS observations of volcanic sulfate aerosol and sulfur dioxide in the stratosphere

Annika Günther[1], Michael Höpfner[1], Björn-Martin Sinnhuber[1], Sabine Griessbach[2], Terry Deshler[3], Thomas von Clarmann[1], and Gabriele Stiller[1]

[1]Karlsruhe Institute of Technology, Institute of Meteorology and Climate Research, Karlsruhe, Germany
[2]Forschungszentrum Jülich GmbH, Institute for Advanced Simulation, Jülich, Germany
[3]University of Wyoming, Department of Atmospheric Science, Laramie, Wyoming, USA

*Correspondence to*: Annika Günther (annika.guenther@kit.edu)

**Abstract**

Volcanic eruptions can increase the stratospheric sulfur loading by orders of magnitude above the background level and are the most important source of variability of stratospheric sulfur. We present a set of vertical profiles of sulfate aerosol volume densities and derived liquid-phase $H_2SO_4$ (sulfuric acid) mole-fractions for 2005–2012, retrieved from infrared limb emission measurements performed with the Michelson Interferometer for Passive Atmospheric Sounding (MIPAS) on board of the Environmental Satellite (Envisat). Relative to balloon-borne in situ measurements of aerosol at Laramie, Wyoming, the MIPAS aerosol data have a positive bias that has been corrected, based on the observed differences to the in situ data. We investigate the production of stratospheric sulfate aerosol from volcanically emitted $SO_2$ for two case studies: the eruptions of Kasatochi in 2008 and Sarychev in 2009, which both occurred in the Northern hemisphere mid-latitudes during boreal summer. With the help of chemical transport model (CTM) simulations for the two volcanic eruptions we show that the MIPAS sulfate aerosol and $SO_2$ data are qualitatively and quantitatively consistent to each other. Further, we demonstrate that the lifetime of $SO_2$ is well explained by its oxidation by hydroxyl radicals (OH). While sedimentation of sulfate aerosol plays a role, we find that the long-term decay of stratospheric sulfur after these volcanic eruptions in mid-latitudes is mainly controlled by transport via the Brewer–Dobson circulation. Sulfur emitted by the two mid-latitude volcanoes resides mostly north of 30° N at altitudes of ~10–16 km, while at higher altitudes (~18–22 km) part of the volcanic sulfur is transported towards the equator where it is lifted into the stratospheric "overworld" and can further be transported into both hemispheres.

## 1 Introduction

Aerosol particles are omnipresent in the atmosphere and can affect climate, air quality, and atmospheric chemistry. In the stratosphere, aerosol particles are mainly composed of sulfuric acid ($H_2SO_4$) and water ($H_2O$) (Kremser et al., 2016; Thomason and Peter, 2006), though organic material has been shown to also play a significant role in the upper troposphere and lower stratosphere ( Yu et al., 2016; Murphy et al., 2014). Stratospheric sulfate aerosol has the potential to directly lower surface temperatures by backscattering parts of the incoming solar radiation. Estimates of the amount of stratospheric

aerosol, and their evolution with time, are therefore important for climate change modelling studies. Increased interest in stratospheric sulfate aerosol is also connected to its potential use in climate engineering schemes (e.g. Niemeier and Timmreck, 2015; Rasch et al., 2008). The negative radiative effect of stratospheric aerosol (Andersson et al., 2015; Brühl et al., 2015; Ridley et al., 2014; Santer et al., 2014; Solomon et al., 2011; Vernier et al., 2011) is discussed to be one of the causes for the global warming "hiatus" experienced during the first decade of this century (Haywood et al., 2014; Santer et al., 2014; Fyfe et al., 2013). Hofmann et al. (2009) observed an increase in stratospheric aerosol load and suggested that this was due to anthropogenic emissions. Newer studies, however, show that this increase is likely to be connected to a number of small and medium sized volcanic eruptions especially in the tropics (e.g. Neely et al., 2013; Vernier et al., 2011). During the last decade, several volcanoes directly injected sulfur into the stratosphere up to 20 km (Vernier et al., 2011). Ridley et al. (2014) and Andersson et al. (2015) emphasise the importance of volcanic aerosol in the lowermost stratosphere in mid- and high-latitudes on the total volcanic aerosol forcing during the last decade. Their studies show that stratospheric altitudes below ~15 km (380 K isentrope), which are not represented in many of the aerosol data sets, need to be taken into consideration when studying the global radiative forcing generated by volcanic eruptions in the extra-tropics.

The main source gases of stratospheric sulfate aerosol during background / non-volcanic conditions are sulfur dioxide ($SO_2$) and carbonyl sulfide (OCS). Due to the longer lifetime of OCS compared to $SO_2$, carbonyl sulfide has a relatively high flux across the tropical tropopause layer (TTL), which is its main entry pathway into the stratosphere. Crutzen (1976) first stated the essential role of OCS for stratospheric aerosol. Chin and Davis (1995), Thomason and Peter (2006), Brühl et al. (2012), and Sheng et al. (2015), agree on a major contribution of OCS to the formation of stratospheric sulfate aerosol. However, the magnitude to which OCS contributes to the stratospheric aerosol loading during background conditions is still under discussion. By emitting $SO_2$, volcanic eruptions are the dominant source for stratospheric $SO_2$ (direct) and sulfate aerosol (indirect) under non-background conditions and cause most of the variability in the stratospheric sulfur loading.

When analysing the vertical extent of $SO_2$ and aerosol plumes after volcanic eruptions and their transport at different altitudes, vertically resolved observations are needed. These are available from satellite-borne limb measurements, such as the Michelson Interferometer for Passive Atmospheric Sounding (MIPAS; Fischer et al, 2008), an instrument that was operational on Envisat (Environmental Satellite). From 2002 to 2012 the instrument provided limb emission measurements in the infrared spectral region. From MIPAS several data sets of trace gas species that are relevant to study the stratospheric sulfur loading are available. These are volume mixing ratios (VMRs) of OCS (Glatthor et al., 2015 and 2017) and $SO_2$ (Höpfner et al., 2013 and 2015). Here, we present a new data set of sulfate aerosol volume densities (AVDs) retrieved from MIPAS measurements, and corresponding $H_2SO_4$ volume mixing ratios. The MIPAS aerosol and $SO_2$ data are tested for consistency using chemical transport model (CTM) simulations in a case study on two volcanic eruptions. Analyses are presented in terms of mass and transport pattern.

In Sect. 2 we first provide basic information on MIPAS, the MIPAS $SO_2$ data set, and balloon-borne in situ data of aerosol volume densities used in this study. This is followed by a short description of the CTM and our model implementations. This paper has several purposes, which are addressed in the subsequent sections. We introduce a new data

set of aerosol volume densities, retrieved from MIPAS measurements in Sect. 3, and compare the data to independent measurements of aerosols. We further study the distribution of MIPAS sulfate aerosol (as VMRs) in the period 2005 to 2012 and compare it to MIPAS $SO_2$. In Sect. 4 we perform a case study for two of the largest volcanic eruptions of the last decade in Northern hemisphere mid-latitudes, which were measured by MIPAS. The volcanoes are Kasatochi (52.2° N/175.5° W) that erupted in August 2008, and Sarychev (48.1° N/153.2° E), which erupted in June 2009. In the case study we analyse MIPAS observations of $SO_2$ and stratospheric sulfate aerosol in comparison to CTM simulations, and study the sulfur mass contained in $SO_2$ and sulfate aerosol, together with the transport of their volcanic plumes. Finally, in Sect. 5, we draw conclusions on the general consistency between the MIPAS $SO_2$ and the new MIPAS sulfate aerosol data set, in combination with our model results in the case of the two volcanic eruptions, and give a short summary of our findings.

## 2 Available observational data sets and model description

### 2.1 MIPAS

#### 2.1.1 Instrument

MIPAS (Fischer et al., 2008) is an infrared (IR) limb emission sounder that was operated on ESAs (European Space Agency) satellite Envisat. The Fourier transform spectrometer measured high-resolution spectra emitted by the constituents of the atmosphere in the thermal IR, in the region 685 to 2,410 $cm^{-1}$ (ESA, 2000). The instrument operated from July 2002 to April 2012, separated in two measurement periods. Here we concentrate on the data from the second and longer measurement period (January 2005–April 2012), as the major mid-latitudinal volcanic eruptions between 2002–2012 occurred during this period. Furthermore, this measurement period is characterised by an improved vertical resolution, especially in the altitude region of the upper troposphere and lower stratosphere. During this period radiance profiles from 7 to 72 km altitude were measured, with an unapodised spectral resolution of 0.0625 $cm^{-1}$, a latitudinal distance of 420 km between two subsequent limb scans, and a vertical sampling step of 1.5 km in the upper troposphere / lower stratosphere region. Installed on a sun-synchronous polar orbiting satellite at an altitude of about 800 km, MIPAS delivered data at around 10am and 10pm, local time. For the retrieval of sulfate aerosol volume densities described in this paper, MIPAS level 1b calibrated radiances Version 5 were used, as provided by ESA.

#### 2.1.2 The $SO_2$ data set

In this study we use the MIPAS $SO_2$ data set as described by Höpfner et al. (2015). Error estimations and a validation of the $SO_2$ data set by comparison with satellite data from the Atmospheric Chemistry Experiment Fourier Transform Spectrometer (ACE-FTS) and other available $SO_2$ observations are provided by Höpfner et al. (2015). Single $SO_2$ profiles of this data set have a total estimated error of around 70–100 pptv and a vertical resolution of 3–5 km. The MIPAS data are shown to be consistent with independent measurements from several aircraft campaigns within ± 50 pptv. With respect to

satellite-borne data from ACE-FTS, MIPAS $SO_2$ shows an altitude-dependent bias between -20 to +50 pptv during volcanically perturbed periods. For background conditions this bias lies between -10 to +20 pptv in the altitude region 10–20 km. For the analysis of volcanically enhanced periods it is necessary to stress complications of the MIPAS $SO_2$ data up to a few weeks directly after the eruption. Under these conditions the total mass of $SO_2$ was found to be strongly underestimated, especially due to aerosol-related sampling artefacts (Höpfner et al., 2015). The study by Höpfner et al. (2015) comprises a data set of volcanically emitted $SO_2$ masses for 30 volcanic eruptions, as observed by MIPAS.

## 2.2 Aerosol in situ measurements from Laramie, Wyoming

To validate the new MIPAS aerosol data set described in Sect. 3, we use aerosol volume density profiles that were derived from in situ measurements of stratospheric aerosol above Laramie, Wyoming (41° N, 105° W) (Deshler et al., 2003). Size resolved aerosol concentration measurements from the surface to approximately 30 km altitude were made with balloon-borne optical aerosol counters, which were developed and operated by the University of Wyoming. Measurements usually occurred between 6 and 9am local time, with measurement frequency varying from monthly to bi-monthly. Data are available from 1971 to present. Over this time period three different primary instrument types were used. The most recent type (laser particle counters, LPCs) of the three instrument types was used first in 2006, became the standard Laramie instrument in 2008, and was, as an example, also flown on quasi-Lagrangian balloons in Antarctica in 2010 (Ward et al., 2014). While the transition from the first instrument to the second was documented in Deshler et al. (2003), a similar study comparing the third Wyoming instrument with the second instrument is a work in progress. For the MIPAS validation, measurements from the second and third Wyoming instruments were available. The positive bias of MIPAS aerosol volumes from the in situ measurements was generally consistent between MIPAS and both of the Wyoming instruments above 20 km. Below 20 km the in situ measurements diverged from each other, with the second instrument indicating higher volumes than the LPC (third instrument), and at times higher than MIPAS. Based on these comparisons with both instruments the Wyoming measurements to be used were confined to those made with the LPC because it permitted the simplest altitude dependent de-biasing function for the MIPAS aerosol volume densities. The LPC measures particles with radii > 0.08–4.2 µm in eight size classes. Deriving geophysical quantities from the size resolved aerosol concentration measurements requires fitting a size distribution to the in situ data. In the past this was done by fitting either a unimodal or bimodal lognormal size distribution to a subset of the measurements. The final size distribution parameters selected are those from that subset of the measurements which minimises the root mean square error when the fitted distribution is compared to all the measurements. This approach is transitioning to a new approach which modifies the nominal in situ aerosol sizes based on laboratory measurements of the aerosol counting efficiency. The counting efficiency at each size is then included in a search of the lognormal parameter space for the lognormal coefficients which minimise the error of the fitted distribution, coupled with the counting efficiency, compared to the measurements. In our study we use the volume density profiles that are derived from the fitted lognormal size distributions (unimodal or bimodal, following the new retrieval approach) to the measurements. The precision of these volume estimates is the same as the old method, ±40 % (Deshler et al., 2003). The

change in the method the fitting parameters are derived is the subject of a paper to be submitted soon. The impact on size distributions from the LPC measurements is not large.

## 2.3 Chemical transport model

The chemical transport model (CTM) used in our study (e.g. Sinnhuber et al., 2003; Kiesewetter et al., 2010) is forced by temperature, wind fields, and diabatic heating rates from the ERA-Interim reanalysis (Dee et al., 2011). The model uses isentropes as vertical coordinates. Horizontal transport on levels of constant potential temperature is derived from the wind fields, while vertical transport is calculated using the diabatic heating rates. The CTM employs the second order moments advection scheme by Prather (1986). The model domain covers 29 isentropic levels between 330 and 2,700 K (~10–55 km), with a horizontal resolution of about 2.5° latitude x 3.75° longitude (Gaussian latitude grid).

As part of this study, a sulfur module has been implemented, including OCS, $SO_2$ and $H_2SO_4$ as advected tracers. In the sulfur module no distinction between tropospheric and stratospheric air is implemented. In the presented simulations, we consider volcanic $SO_2$ from one volcanic eruption as the only sulfur source per simulation, in order to study the $SO_2$ and sulfate aerosol after the eruptions of Kasatochi in 2008 and Sarychev in 2009 individually. Therefore, OCS is not considered as sulfur source in the simulations presented in this work. The bottom boundary concentrations and initial fields are set to zero for $SO_2$ and $H_2SO_4$. The volcanic $SO_2$ is injected instantaneously into the model column of the volcano. $SO_2$ masses are injected into three altitude ranges (see Sect. 4.1, Table 1), consistent with MIPAS $SO_2$ observations, and $SO_2$ is uniformly distributed to the air mass within each altitude range. The sulfur released from volcanic $SO_2$ reacts with OH (hydroxyl radical) to form $H_2SO_4$. This is the only chemical reaction considered in the simulations presented in this study. As the interim product sulfur trioxide ($SO_3$), formed during the oxidation of $SO_2$, combines rapidly with water vapour to form sulfuric acid, it is not explicitly considered in the model. The removal processes of the species are advective transport out of the model domain into the troposphere, decay of $SO_2$ by its reaction with OH, and gravitational settling of sulfate aerosol. An OH climatology of daily distributions was derived in a full chemistry run of the CTM (2003–2006), and for reaction rates the recommendations from JPL (Jet Propulsion Laboratory) are used (Burkholder et al., 2015). For the sedimentation of sulfate aerosol equilibrium partitioning between gas- and liquid-phase is assumed for $H_2SO_4$ (Ayers et al., 1980) at each simulation time step. To determine the terminal velocity for the part of the sulfate aerosol that settles, velocity calculations follow the approach suggested by Jacobson (1999). In the simulations presented here, the aerosol radius is fixed to one effective settling radius. The solution density of the aerosol is calculated online from the fraction of liquid-phase $H_2SO_4$ in the binary solution of the $H_2SO_4$–$H_2O$ aerosol. The sulfur scheme hydrates the sulfate aerosol based on ambient water vapour loading, which is assumed to be 4.5 ppmv. Sedimentation transports sulfate aerosol into the grid box below, or finally out of the model domain. All model results shown for $H_2SO_4$ only consist of the sulfate aerosol droplets, as the MIPAS measurements do not consider gas-phase $H_2SO_4$.

The model is run for 365 d per simulation, with a time step of 30 min and tracer fields are written out daily at 12 UTC. For the eruption of Kasatochi (7 Aug 2008) the individual runs are started on the 31 Jul 2008, and for the eruption of Sarychev (12 Jun 2009) all runs are started on the 31 May 2009. As the relevant initial trace-gas fields are set to zero and the model is driven by ERA-Interim reanalysis data, which are updated every six hours, no long spin up time is needed. Per volcano four simulations were made that differ concerning the particle size of sulfate aerosol. Simulations were made with constant radii of 0.1, 0.5 and 1 µm, and without sedimentation. In the atmosphere the radius of sulfate aerosol varies (Deshler et al., 2003 and 2008). Nevertheless, for simplification, we use a constant "effective sedimentation radius" to determine the terminal fall velocity, which we consider to be the average settling speed of aerosol particles of different radii. In our simple sulfur scheme, no scavenging of $SO_2$ or $H_2SO_4$ by clouds is considered in the model. This would be confined mostly to tropospheric altitudes and in our study region ($\geq$ 10 km) especially to tropical latitudes. Washout by precipitation might play a role there but it is expected to have a minor effect on our study, as we analyse the sulfur that remains in the atmosphere (above ~10 km) after the first weeks following the volcanic eruptions. Furthermore, no nucleation or growth processes of sulfate aerosol are considered.

## 3 The new MIPAS aerosol data set

### 3.1 Aerosol retrieval from MIPAS limb-spectra

In previous analyses of mid-infrared observations by MIPAS-B (the balloon-borne predecessor of the MIPAS satellite instrument; Friedl-Vallon et al., 2004) and MIPAS/Envisat (MIPAS instrument on the satellite Envisat, generally referred to as "MIPAS" throughout the present work) it has been demonstrated that the limb radiances due to particles have two major contributing terms: (1) the thermal emission of the particles, and (2) the scattered radiation from the atmosphere and Earth's surface from below the tangent point (Höpfner et al., 2002 and 2006). The relative weights of these contributions differ with particle size and wavenumber. For particles sufficiently small compared to the wavelength (d < ~1 µm in the mid-IR; Höpfner, 2004), the scattered contribution can be neglected such that only the thermal emission remains as major source of IR radiation. In this wavenumber length regime the radiance only depends on the total aerosol volume density. Typical sizes of the stratospheric aerosol layer particles are less than 1 µm in case of background and enhanced conditions due to medium sized volcanic eruptions (e.g. Deshler et al., 2003). Thus, our retrieval target is the altitude profile of volume densities, derived from each set of calibrated MIPAS limb-scan spectra.

For this study we have concentrated on the second MIPAS measurement period between January 2005 and April 2012. The retrieval model used is the KOPRA/KOPRAFIT (Karlsruhe Optimized and Precise Radiative transfer Algorithm) suite, allowing to directly retrieve aerosol parameters from observed radiances by coupling a Mie-model with the line-by-line radiative transfer scheme (Stiller et al., 2002; Höpfner et al., 2002 and 2006). For aerosol composition we assume a 75 percent by weight (75 wt%) $H_2SO_4$–$H_2O$ solution, as the stratospheric sulfuric aerosol composition typically varies between around 70 and 80 %, as obtained by equilibrium calculations (Carslaw et al., 1995) and observations (e.g. Doeringer et al.,

2012). Kleinschmitt et al. (2017), calculating aerosol optical properties, Kremser et al. (2016), calculating sulfur fluxes, and Gao et al. (2007), calculating atmospheric volcanic aerosol loadings, also use a 75 wt% $H_2SO_4$–$H_2O$ composition in their studies. The imaginary parts of various refractive index data sets in the mid-IR are displayed in Fig. 1. Here the used optical constants by Niedziela et al. (1999) for 75 wt% and 230 K (bold red line) are compared to data at other concentrations and temperatures by Niedziela et al. (1999) (upper panel), and Myhre et al. (2003) (lower panel). This particular data set has been chosen because in an evaluation of optical constants for sulfuric acid, Wagner et al. (2003) have found those data sets to be best consistent with observations in the aerosol chamber AIDA (Aerosol Interactions and Dynamics in the Atmosphere). The spectral range selected for the retrieval (1,216.5–1,219.5 cm$^{-1}$) is situated at the long wavelength end of MIPAS band B as indicated by the two vertical lines in Fig. 1. It lies within one of the atmospheric windows as can be seen by comparison to the limb-transmission curve (light grey) in Fig. 1. We have not chosen the windows at around 830 cm$^{-1}$ and 950 cm$^{-1}$ since at 1,220 cm$^{-1}$ the absorption by $H_2SO_4$ droplets is higher and the relative difference between the various sets of refractive indices is smaller.

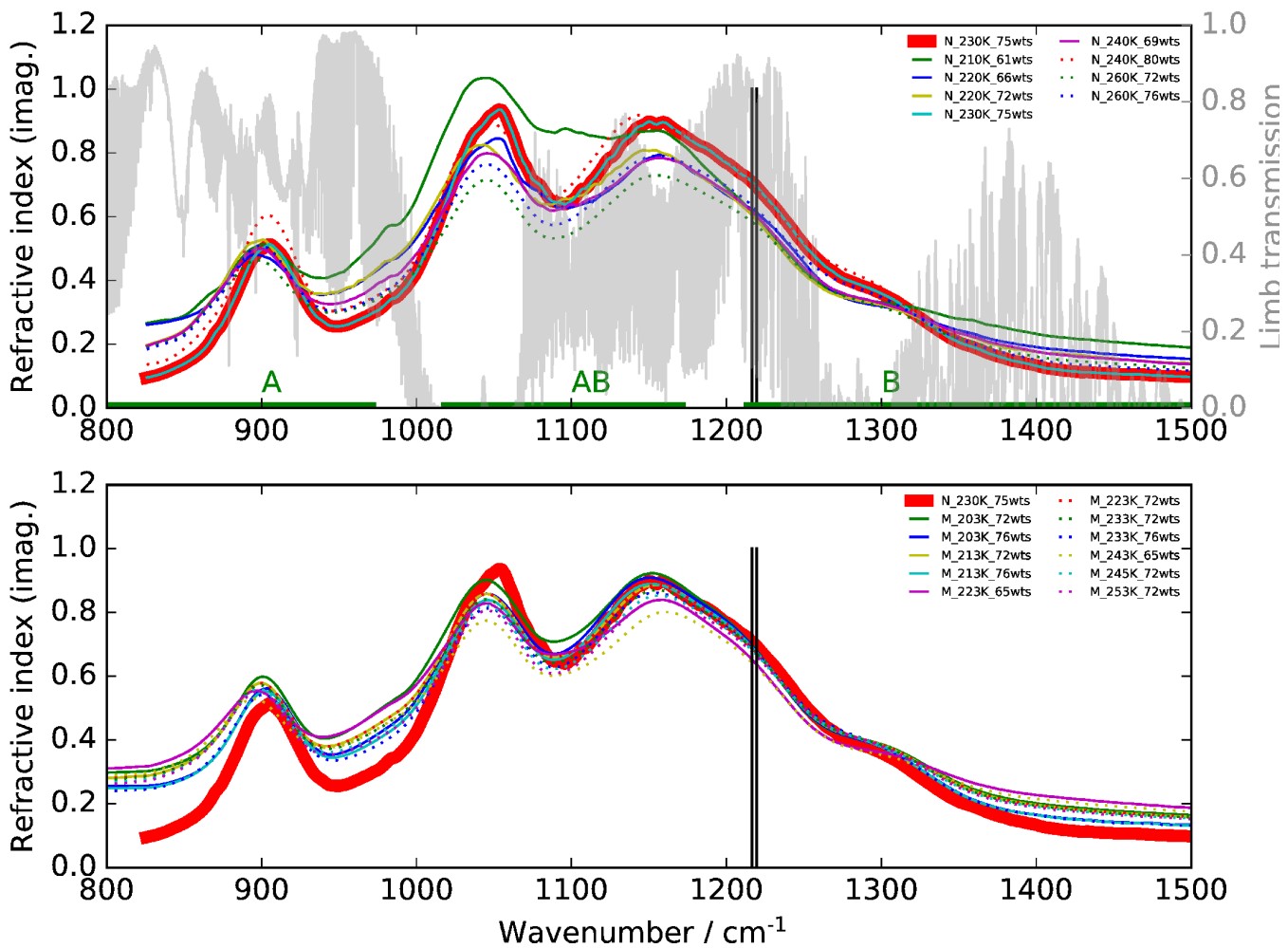

**Figure 1.** Imaginary parts of refractive indices for aqueous $H_2SO_4$ solutions of different concentrations (wts: weight % $H_2SO_4$) and different temperatures in the mid-infrared region. **(Top)** thin solid and dotted curves: Niedziela et al (1999), **(bottom)** Myhre et al. (2003). The bold red line indicates the data set, and the two vertical lines the spectral window used in this study. A simulated limb transmission spectrum for 10 km tangent altitude for standard mid-latitude conditions is additionally plotted in the top row as well as the wavenumbers covered by MIPAS bands A, AB and B.

The retrieval has been set up as a multiparameter nonlinear least-squares fit of the calculated to the measured limb radiances of entire limb-scans (e.g. von Clarmann et al., 2003). Besides the target parameter, namely sulfate aerosol volume densities, further atmospheric fit-parameters of the retrieval are vertical profiles of spectrally interfering trace gases methane ($CH_4$), $H_2O$, ozone ($O_3$), and nitric acid ($HNO_3$). While zero initial guess profiles have been used for the aerosol volume densities, results from the IMK routine processing are taken for the trace gases (von Clarmann et al., 2009). As the atmospheric parameters are represented at denser altitude levels (1 km) than the vertical field-of-view (~3 km) and the vertical tangent point spacing (1.5 km) of MIPAS, constraints on the smoothness of the profile shape are introduced by

regularisation (Tikhonov, 1963; Steck, 2002). The retrieval of aerosol volume density is restricted to altitudes up to 33 km and the regularisation strength has been adjusted such that its resulting vertical resolution is around 3 to 4 km. To cover instrumental uncertainties, a spectral shift parameter and a radiance offset, constant over all wavenumbers and tangent altitudes, have been retrieved simultaneously to the atmospheric quantities. For the analysis in this paper, only data at altitudes with averaging kernel diagonal elements larger than 0.05, which refer to altitudes at least 1 km above the lowest tangent height, are used.

An overview of the leading error components is presented in Fig. 2, with the assumed parameter uncertainties listed in the caption. The error contributions are estimated from a subset of a few hundred single cases by sensitivity studies using perturbed parameters or, in case of spectral noise, directly from the retrieval diagnostics. The total error changes with altitude from around 20 % ($0.09 \, \mu m^3 \, cm^{-3}$) at 10 km up to over 40 % ($0.005 \, \mu m^3 \, cm^{-3}$) at 30 km. It is dominated by the uncertainty of the optical constants resulting in 10–20 % error, followed by tangent pointing knowledge with 5–15 %. The error component resulting from spectral noise is rather constant with altitude in absolute terms of volume density and amounts to about $0.01 \, \mu m^3 \, cm^{-3}$. Other instrumental errors that have been investigated but are not listed in Fig. 2 are uncertainties due to the knowledge of the instrumental line shape, and radiometric gain and offset calibration error. In the estimation of the radiative error no radiometric offset variation with tangent altitude was considered, and, thus, not compensated for by the retrieval approach. However, a tangent altitude dependent radiometric offset error caused, e.g., by straylight in the instrument cannot be excluded (López-Puertas et al., 2009). We have not handled this uncertainty in the framework of error estimation but we have tried instead to compensate for it through a de-biasing of the data set based on validation with in situ observations as described in Sect. 3.2.

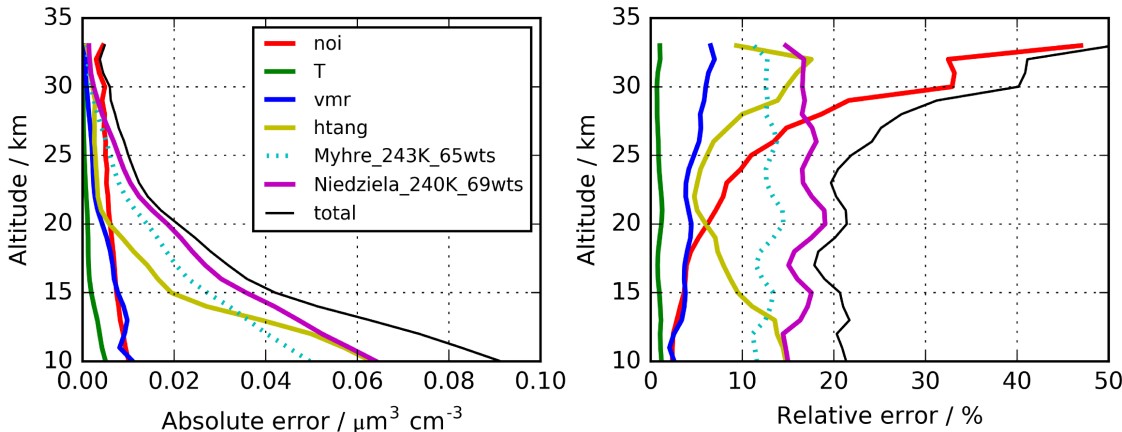

**Figure 2.** Altitude dependent estimated errors for the retrieval of $H_2SO_4$–$H_2O$ aerosol volume densities. Solid lines indicate the uncertainties used to calculate the 'total' error. Indicated errors are: 'noi': single scan spectral noise, 'T': temperature uncertainty 2 K, 'vmr': 10 % uncertainty of volume mixing ratios of interfering gases, 'htang': 300 m tangent altitude uncertainty, 'Niedziela_240K_60wts': use of optical constants by Niedziela et al. (1999) for 240 K and 69 wt% instead of Niedziela et al. (1999) for 230 K and 75 wt% $H_2SO_4$. The dotted curves ('Myhre_243K_65wts') show the results when the optical constants of Myhre et al. (2003) for a temperature of 243 K and a concentration of 65 wt% are used instead of those by Niedziela et al. (1999) (T=230 K and 75 wt%).

Prior to the retrieval, a deselection of spectra affected by clouds has been performed via application of an established cloud filter method for MIPAS by Spang et al. (2004). To sort out optically thick clouds, but not all aerosol-affected spectra, this cloud filter has been applied with a cloud index limit of 1.7. Due to this loose setting of the cloud-filter, artefacts caused, e.g., by thin cirrus clouds, polar stratospheric clouds (PSCs) or volcanic ash remain in the data set, which are all attributed to the retrieved 75 wt% $H_2SO_4$–$H_2O$ aerosol volume density. Thus, further filtering of affected profiles has been necessary after completing the retrieval.

Two distinct features of strong enhancements with an annual cycle show up in the unfiltered data set. The first feature is due to strong enhancements in the presence of PSCs at the winter pole. To deselect PSC-affected profiles a filter is applied when temperatures in the altitude range 17–23 km drop below a threshold of 195 K polewards of 40°, for the Northern hemisphere from 15 Nov–15 Apr, and for the Southern hemisphere from 1 Apr–30 Nov. This temperature represents the nitric acid trihydrate (NAT) existence temperature at around 20 km under typical stratospheric conditions for nitric acid and water vapour.

The second feature is assumed to be induced by thin cirrus clouds. It is present mainly in the tropics, at around 25° S–25° N, at altitudes between about 13–21 km. It reaches highest altitudes and is most intense above and in the vicinity of continents, and above the western Pacific. The vertical extent is smallest in boreal summer, and its vertical gradient towards lower aerosol volume densities is relatively strong, with no upward transport being observed. Bot the location in the tropics, in regions of strong vertical motions and convective clouds, and the relatively sharp decrease at higher altitudes towards

increasing temperatures, suggest that it is connected to the influence of ice particles. The ice-filter for MIPAS data by Griessbach et al. (2016) is applied on all retrieved MIPAS aerosol profiles to reduce the effect of spectra influenced by ice in the present data set. Their method consists of two steps to detect whether MIPAS spectra are influenced by aerosols, ice, clouds, ashes, or a clear sky (Griessbach et al., 2014 and 2016). First, aerosols and clouds are identified, using a spectral

window region that is sensitive to aerosols and clouds. Then ice clouds and aerosols are discriminated, using information from spectral windows with contrasting behaviour for ice and aerosols. In our data set, we consider only retrieved values starting 4 km above the altitude of the uppermost spectrum that was flagged to have been influenced by ice. Further, the ash filter for MIPAS spectra by Griessbach et al. (2014), based on an ash detection threshold function, is applied in the same way as the ice filter, to filter out volcanic ash and mineral dust.

**3.2 Validation and bias correction**

To validate the new data set, we compare the profiles of MIPAS aerosol volume density to in situ balloon measurements (Deshler et al., 2003). In situ measurements were carried out with laser based aerosol spectrometers from Laramie, Wyoming (41° N/105° W), between 6 and 9am, local time. In Fig. 3, profiles of the balloon measurements are shown. In comparison, MIPAS mean aerosol volume density profiles are presented, selected from a restricted area around Laramie, together with

their standard errors (Fig. 3), which show the variability of the underlying profiles. MIPAS profiles are chosen for the day of the balloon flight within a 5° latitude x 10° longitude distance around Laramie. Further, in Fig. 4a we show the mean over the profiles that were retrieved from LPC measurements, as shown in Fig. 3 (excluding the 28 Jul 2011) and the corresponding MIPAS profiles, together with the absolute (b) and relative difference (c) of the mean MIPAS profile to the mean profile from balloon measurements. In Fig. 4a the measurement uncertainties of the mean profiles are presented, and in

Fig. 4b the uncertainty of the bias (difference between the mean in situ and mean MIPAS profile) is shown. The profile on the 28 Jul 2011 shows large differences between MIPAS and the in situ data and the strongest vertical variability of all in situ profiles at low altitudes (below ~18 km), possibly due to the Nabro eruption (12 Jun 2011). Hence, this profile is excluded from the calculation of the mean profiles shown in Fig. 4.

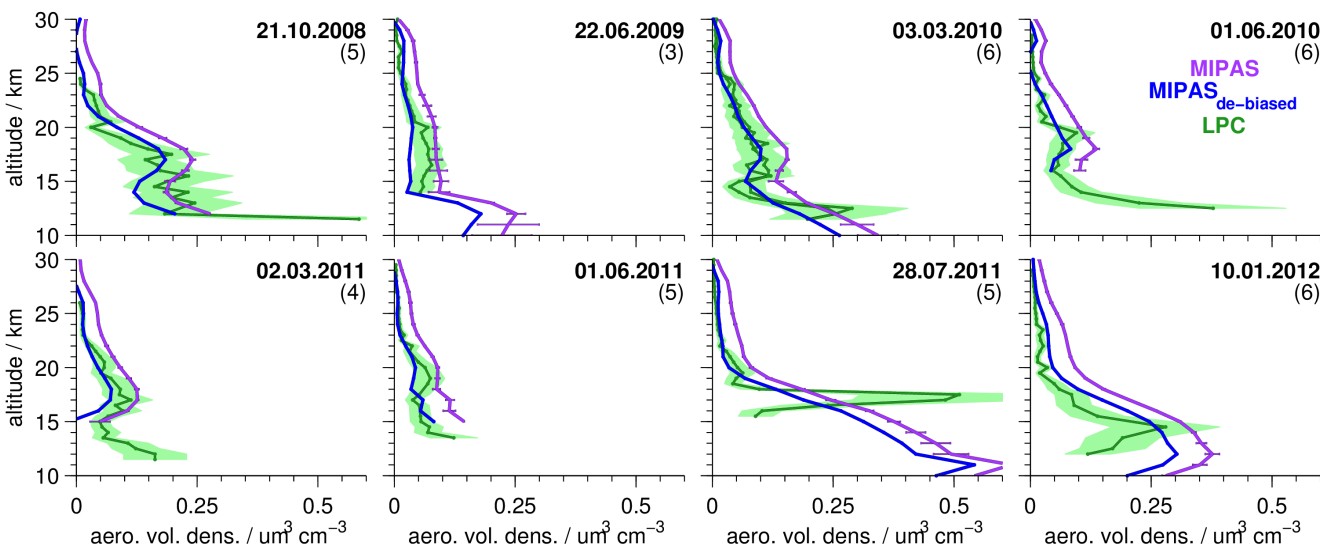

**Figure 3.** Profiles of aerosol volume densities from in situ and MIPAS satellite measurements. In situ data are balloon-borne data above the lapse rate tropopause for Laramie, Wyoming (41° N/105° W), measured by laser particle counters. Two instruments were flown on the 3 Mar 2010. The 40 % errors of the in situ data are shown as green shaded areas. MIPAS satellite data in purple (original data) and blue (de-biased data). On all MIPAS data the ice filter by Griessbach et al. (2016) and ash filter by Griessbach et al. (2014), and our PSC filter have been applied. MIPAS profiles are averages of all available profiles for the indicated day, averaged in a ± 5° lat x 10° lon area. In brackets are the number of MIPAS profiles included in the average. For the non-de-biased MIPAS values the 1-sigma standard errors of the mean profiles are shown.

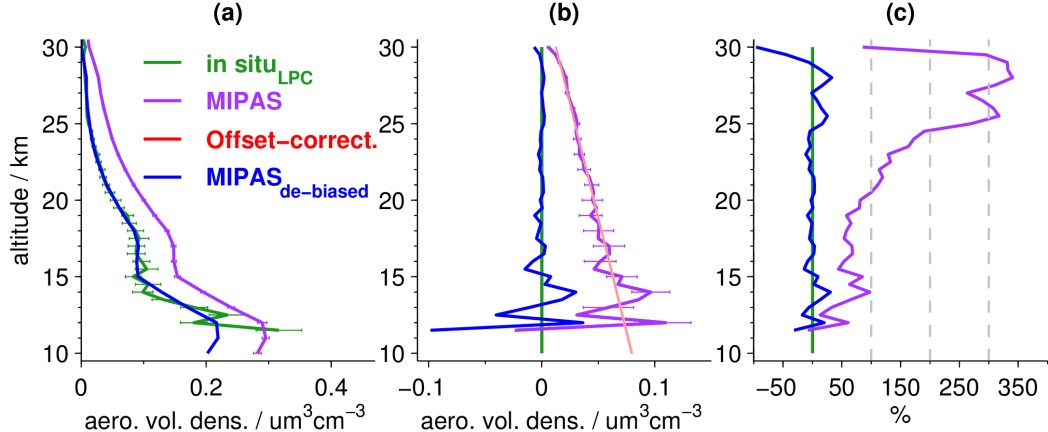

**Figure 4.** Aerosol volume densities by MIPAS and in situ data from Laramie, Wyoming. **(a)** mean profiles for data in Fig. 3 (excluding the 28 Jul 2011), and their 1-sigma errors, based on the estimated measurement uncertainties. In blue are de-biased MIPAS values, in purple the original data. **(b)** absolute and **(c)** relative differences of MIPAS to LPC in situ data. Differences are calculated for the mean profiles shown in (a). In (b) the uncertainties of the absolute differences between in situ and MIPAS data are shown (horizontal purple lines; error propagation based on MIPAS and in situ errors presented in (a)). Red line: linear fit to the profile of absolute differences (purple line) between 18 and 30 km, used to reduce the bias to the LPC in situ measurements (light red area: 1-sigma uncertainty of the linear fit).

Generally, the aerosol volume densities (Fig. 3 and 4) are highest in the lower stratosphere and then decrease towards zero at higher altitudes. As the balloon data have a higher vertical resolution, and the retrieval process for MIPAS profiles includes smoothing, the in situ data show finer structures. Compared to the balloon data, the original MIPAS aerosol volume densities show a positive bias in most profiles (Fig. 3) as well as in the mean profile (Fig. 4). This is most easily detectable at higher altitudes where profiles are relatively smooth. The offset amplifies towards lower altitudes (Fig. 4b). Aiming on a reduction of this positive offset, a height-dependent de-biasing is performed on all single MIPAS profiles. The de-biasing is based on the in situ measurements carried out with laser based particle counters. MIPAS profiles show a consistent variation with height, compared to the LPC measurements. An additive linear de-biasing is applied, rather than a multiplicative correction, as the offset is expected to be caused by an altitude-dependent additive stray light error in the radiances (see Sect. 3.1). The de-biasing is based on the absolute differences between the aerosol volume densities of the mean MIPAS and in situ profiles (Fig. 4b, purple solid profile) at 18–30 km where profiles show weak variability and a relatively low uncertainty of the bias. A regression line (Fig. 4b, red line) to the profile of absolute differences represents the vertically resolved values of the de-biasing function, which are subtracted from each MIPAS profile during offset-correction. A narrow red shaded area indicates the uncertainty of the bias-correction and has been evaluated using generalised Gaussian error propagation of the uncertainties of the slope and the intercept of the regression line. No weighting of the data points by their inverse error variances was applied in the calculation of mean in situ and MIPAS profiles. This method has been chosen in order to avoid representativeness problems, as the error variances correlate with the aerosol loading of the atmosphere and

would thus cause a sampling artefact in the estimated bias and offset-correction. At lower altitudes where profiles show more variability, both vertically and between the in situ and MIPAS profile, the linear fit also suits well. The uncertainty of the bias (Fig. 4b) shows that the positive bias is not random, as the spread is rather low and uncertainty limits are noticeably distant from zero. The mean de-biased MIPAS profile (Fig. 4a) matches the in situ data and lies mostly in the range of the uncertainties of the mean in situ profile. Further, the absolute and relative differences to the balloon data are reduced significantly (Fig. 4b and c). Percentage differences are mostly below ± 25 %. For the non-de-biased profile, at altitudes above around 20 km, percentage differences increase strongly due to very low aerosol volume densities, while at lower altitudes percentage differences are below about 100 %. By excluding the in situ and MIPAS profiles measured on 28 Jul 2011 from the calculation of the mean profiles, the agreement between the measurements is improved in the altitude range below 18~km, while above this altitude changes are marginal, as can be expected from Fig. 3. The de-biasing is therefore not affected by disregarding the observations from this day.

## 3.3 Time series of MIPAS sulfate aerosol and $SO_2$ for 2005 to 2012

To study the distribution of sulfate aerosol as measured by MIPAS from 2005 to 2012, Fig. 5 (left) shows latitudinally resolved time series of liquid-phase $H_2SO_4$ mole-fractions, for various altitudes from 10 to 22 km. From the retrieved aerosol volume densities, the mole-fractions are calculated by assuming all aerosol to be sulfate aerosol with a composition of 75 wt % $H_2SO_4$ and 25 wt% $H_2O$, and an aerosol density of 1,700 kg m$^{-3}$. Strongest variability in the MIPAS sulfate aerosol data is caused by volcanic eruptions. In the Northern hemisphere, strongest signatures of volcanic eruptions are due to the eruptions of Kasatochi (52.2° N/175.5° W) in August 2008, Sarychev (48.1° N/153.2° E) in June 2009, and Nabro (13.4° N/41.7° E) in June 2011, at altitudes from 10 to about 20 km. At low latitudes tropical volcanoes such as Manam (4.1° S/145.0° E) in January 2005, Soufrière Hills (16.7° N/62.2° W) in May 2006, and Rabaul (4.3° S/152.2° E) in October 2006, increase the sulfate aerosol mole-fractions at higher altitudes, above 16 km. The aerosol is lifted upwards with time and the plumes get modulated by the Quasi-Biennial Oscillation in the tropics. A similar pattern of upward motion of the volcanic aerosol from these tropical eruptions has been seen in satellite measurements of aerosol extinction ratios (Vernier et al., 2011). In the Southern hemisphere the eruption of Puyehue-Cordón Caulle (40.6° S/72.1° W) in June 2011 has the strongest impact on the measurements, but is restricted to lower altitudes, below about 14/15 km. During the preceding years the mole-fractions are relatively low in the mid-latitudes of the Southern hemisphere, at 10 to 12 km.

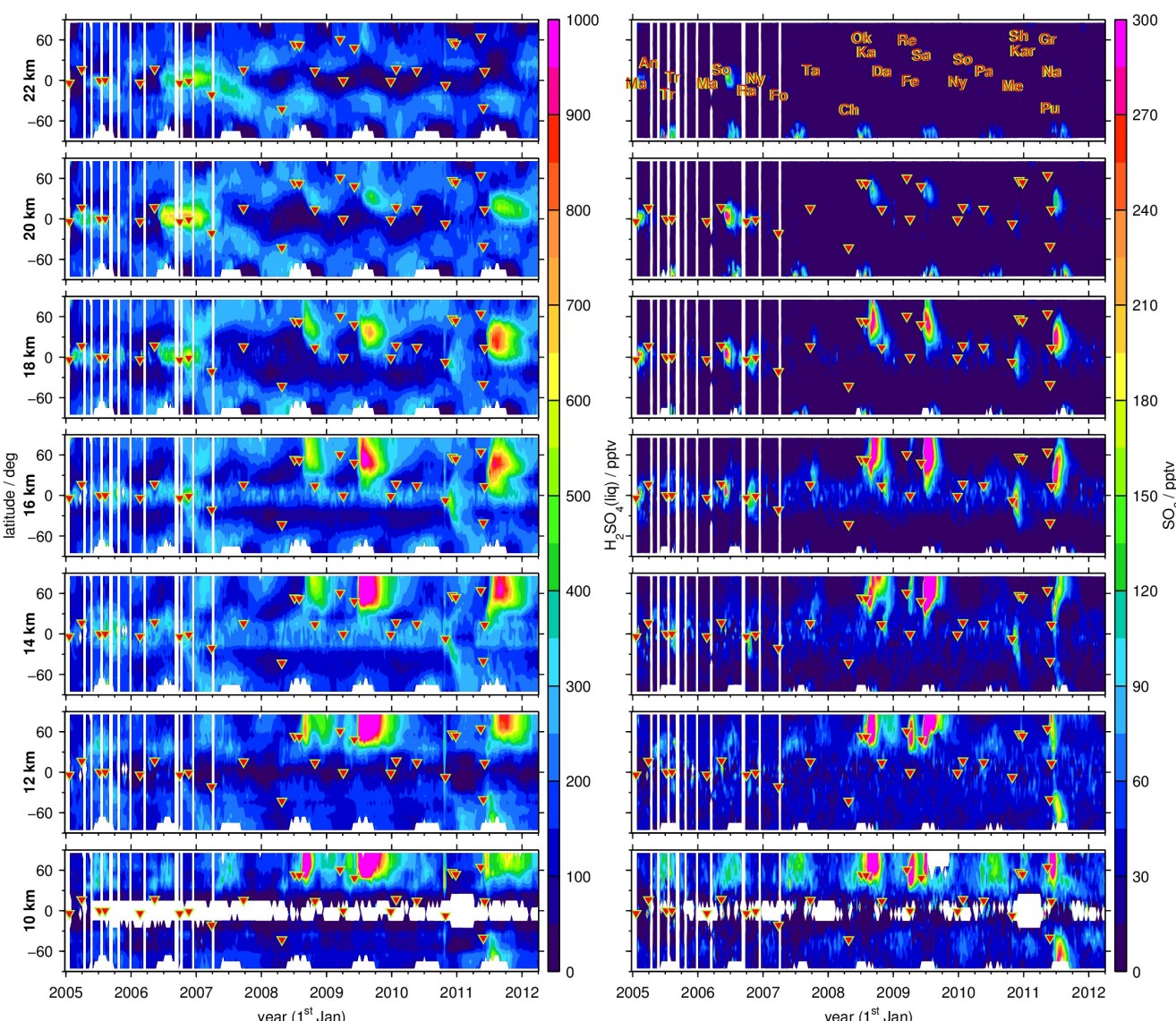

**Figure 5.** Global time series of latitudinally resolved distributions of MIPAS liquid-phase $H_2SO_4$ **(left panels)** and $SO_2$ **(right panels)** volume mixing ratios. Shown are 8-day mean values for 10° latitude bins at different altitudes (10 to 22 km). The colour-code is restricted to 0–1,000 pptv for $H_2SO_4$(liq), and 0–300 pptv for $SO_2$. Values exceeding these limits are assigned with the limiting value, respectively. Volcanic eruptions are indicated by red triangles (together with their abbreviated names in the uppermost right panel; following Höpfner et al., 2015). Abbreviations: An: Anatahan, Ch: Chaiten, Da: Dalaffilla, Fe: Fernandina, Fo: Piton de la Fournaise, Gr: Grímsvötn, Ka: Kasatochi, Kar: Karymsky, Ma: Manam, Me: Merapi, Na: Nabro, Ny: Nyamuragira, Ok: Okmok, Pa: Pacaya, Pu: Puyehue-Cordón Caulle, Ra: Rabaul, Re: Redoubt, Sa: Sarychev, Sh: Shiveluch, So: Soufrière Hills, Ta: Jebel at Tair, Tr: unidentified Tropical Volcano.

The latitudinally resolved time series of sulfate aerosol mole-fractions further reveal different periodic structures, which are not connected to volcanic activities:

1. In polar regions at altitudes above ~16 km, sulfate aerosol mole-fractions decrease strongly in winter to spring. The pattern is more pronounced in the Southern hemisphere. This decrease is connected to the polar vortex, where relatively sulfate aerosol free air is transported downwards. Thomason and Poole (1993) reported on very low observed aerosol levels relative to non-vortex air.

2. In both hemispheres, but primarily in the Southern hemisphere, mole-fractions of liquid-phase $H_2SO_4$ are enhanced at around 20–22 km in the mid-latitudes (and partly the tropics) during boreal / austral winter and spring, respectively. In the stratosphere sulfur is released from OCS mainly in the tropics at altitudes between about 25 and 35 km (Brühl et al., 2012) and the sulfate aerosol that is produced is transported towards mid-latitudes and lower altitudes.

3. In the mid-latitudes of the Northern hemisphere, the sulfate aerosol is increased during boreal summer at around 10–12 km.

4. In the tropics at around 14–16 km aerosol values are elevated, while they are very low below and above these altitudes, unless influenced by volcanic eruptions.

As $SO_2$ is the main precursor for stratospheric sulfate aerosol during volcanically perturbed times, we analyse similarities and discrepancies between the distribution of latitudinally resolved time series of MIPAS $SO_2$ at various altitudes for 2005–2012 (Fig. 5, right), and the new aerosol data (Fig. 5, left). Prominent features seen in the distribution of $SO_2$ mole-fractions from single limb scans are described by Höpfner et al. (2015). It should be noted, that these data are useful mainly for the analysis of enhanced $SO_2$, rather than for background conditions, for which the monthly and zonal mean MIPAS $SO_2$ data set by Höpfner et al. (2013) is more suited. The distributions of $SO_2$ and sulfate aerosol show clear similarities, especially concerning volcanic plumes. Differences in the patterns result mostly from the longer residence time of sulfate aerosol in the stratosphere, compared to $SO_2$. Sulfate aerosol can reside in the stratosphere for several months up to several years, if it is neither being transported back to the troposphere nor evaporated at higher altitudes. On the contrary, $SO_2$ has a stratospheric lifetime of a few weeks. Different point sources, such as volcanic eruptions, can therefore be distinguished more easily in the $SO_2$ measurements than in the aerosol data. Further discrepancies arise from the fact that sulfur is released from $SO_2$ over weeks, during the exponential decay of the latter, and can then be converted into sulfate aerosol. Elevated $SO_2$ amounts are therefore not instantly leading to elevated sulfate aerosol amounts, and the curve of enhanced sulfate aerosol is broader and flatter than for $SO_2$.

In the Northern hemisphere at low altitudes (< 12/13 km) during boreal summer a similar feature of increased VMRs is present in the $SO_2$ as in the sulfate aerosol data (point 3). A closer look at the monthly distribution of $SO_2$ and sulfate aerosol reveals no distinct patterns (for $SO_2$ see Höpfner et al., 2015); enhancements are spread over the entire Northern hemisphere. In Höpfner et al. (2015) this feature could not be confirmed due to a lack of $SO_2$ in situ data. The presence of similar enhancements in the aerosol data supports the hypothesis of the increased sulfur loading at low altitudes in the Northern hemisphere not being a retrieval artefact. Further, elevated values in the tropics at around 14–16 km, as seen in the MIPAS

aerosol (point 4) are also present in the $SO_2$ data. These are localised mostly in continental regions, and the western Pacific, both for MIPAS sulfate aerosol and $SO_2$ (for $SO_2$ see Höpfner et al., 2015).

## 4 Volcanic eruptions of Kasatochi in 2008 and Sarychev in 2009

We present a case study of MIPAS $SO_2$ and sulfate aerosol measurements and CTM model simulations for the two volcanic eruptions of Kasatochi and Sarychev (Kasatochi: 7 Aug 2008, 52.2° N/175.5° W; Sarychev: 12 Jun 2009, 48.1° N/153.2° E). Both volcanoes erupted at Northern hemisphere mid-latitudes during boreal summer. MIPAS satellite measurements are compared to CTM simulations, to study the evolution of the emitted sulfur in terms of conversion from $SO_2$ to sulfate aerosol, and its transport and removal at altitudes between 10 and 22 km. As our intention is to study explicitly the sulfur per volcanic eruption, background values per model simulation are set to zero for both $SO_2$ and $H_2SO_4$, and no other sources than the volcanically emitted $SO_2$ of one volcanic eruption is included.

### 4.1 Sulfur mass in the Northern hemisphere mid- and high-latitudes

In this section we aim at testing the agreement between measured $SO_2$ and liquid-phase $H_2SO_4$ masses, together with modelled data, in terms of the increase and decline of sulfur emitted by the volcanic eruptions of Kasatochi in August 2008 and Sarychev in June 2009, and the influence of the prescribed effective sedimentation radius on the residence time of sulfate aerosol. As we intend to test if the measured aerosol is quantitatively and qualitatively consistent with its measured precursor by comparison with modelled sulfate aerosol, a good agreement between the modelled and measured $SO_2$ masses is essential.

In Table 1 injected $SO_2$ amounts for three altitude regions are given (labelled "present study"), as used for the CTM simulations in the present study, together with comparisons to volcanic $SO_2$ masses from the literature. The upper injection limit for the volcanic emissions in the CTM is set to 19 km. Simulations have been made with varying injected $SO_2$ masses and upper injection altitude limits, intending to achieve good agreement between the modelled and measured $SO_2$ masses (comparisons as in Fig. 6). The data presented here resulted in the best agreement, with comparisons starting approximately one month after the respective eruption (explanation in the following). Due to the limited number of simulations no uncertainties are given for the presented $SO_2$ masses. The main part of $SO_2$ per eruption is emitted into the altitude region from 10 to 18 km, and only few percent of the $SO_2$ masses are injected into altitudes above 18 km. Our best match for Kasatochi is consistent with the lower limit of Höpfner et al. (2015). Höpfner et al. (2015) derived volcanic $SO_2$ masses for three altitude regions from 10–14 km, 14–18 km, and 18–22 km by exponential extrapolation of the MIPAS $SO_2$ masses back to the eruption day. They applied this method as in the first month after the eruption MIPAS underestimates the $SO_2$ (Höpfner et al., 2015). Their method results in relatively large error bars that depend on the time period the fit is based on (Höpfner et al., 2015; presented also in Table 1). For Sarychev, however, our best estimate is smaller than the error limits of the $SO_2$ masses given by Höpfner et al. (2015).

When comparing the $SO_2$ masses from different studies, it has to be pointed out that the $SO_2$ masses are generally not derived for the same altitude regions. Höpfner et al. (2015), Brühl et al. (2015), and the present study are not totally independent from each other, as they are entirely or partly based on the same MIPAS $SO_2$ data by Höpfner et al. (2015). The $SO_2$ masses in our study lie below those of all studies but Brühl et al. (2015) for Kasatochi, and in the range of the other publications for Sarychev. The wide range of $SO_2$ masses in Table 1 shows the difficulties and uncertainties related to the determination of volcanically emitted $SO_2$.

**Table 1.** Volcanically emitted $SO_2$ masses from various publications for Kasatochi in 2008 and Sarychev in 2009. For Höpfner et al. (2015) the given total uncertainty is the sum of the uncertainties per altitude range (Table 3 therein). In the case of Pumphrey et al. (2015) pressure levels are given. These represent the highest pressure used for vertical integration.

| Kasatochi in 2008 | | | Sarychev in 2009 | | |
|---|---|---|---|---|---|
| | SO₂ mass Gg | Height range | | SO₂ mass Gg | Height range |
| Present study | 677 | 10–19 km | Present study | 768 | 10–19 km |
| | 518 | 10–14 km | | 401 | 10–14 km |
| | 124 | 14–18 km | | 362 | 14–18 km |
| | 35 | 18–19 km | | 5 | 18–19 km |
| Höpfner et al. (2015) | 898 (± 221) | 10–22 km | Höpfner et al. (2015) | 1,474 (± 357) | 10–22 km |
| | 645 (± 127) | 10–14 km | | 888 (± 293) | 10–14 km |
| | 210 (± 86) | 14–18 km | | 542 (± 60) | 14–18 km |
| | 43 (± 8) | 18–22 km | | 44 (± 4) | 18–22 km |
| Brühl et al. (2015) | 376 | | Brühl et al. (2015) | 562 | |
| Pumphrey et al. (2015) | 1,350 (± 38) | 215 hPa | Pumphrey et al. (2015) | 571 (± 42) | 147 hPa |
| Clarisse et al. (2012) | 1,600 | | | 1,160 (± 180) | 215 hPa |
| Karagulian et al. (2010) | 1,700 | | Clarisse et al. (2012) | 900 | |
| Prata et al. (2010) | 1,200 | | Haywood et al. (2010) | 1,200 (± 200) | |
| Kristiansen et al. (2010) | 1,000 | > 10 km | Carn et al. (2016) | 1,200 | |
| Krotkov et al. (2010) | 2,200 | | | | |
| Corradini et al. (2010) | 900–2,700 | | | | |
| Thomas et al. (2011) | 1,700 | | | | |
| Carn et al. (2016) | 2,000 | | | | |

Time series of sulfur mass contained in $SO_2$ and sulfate aerosol are shown in Fig. 6 for observations by MIPAS and simulations by the CTM. The sulfur loading is shown for three altitude regions, from 10.5–14.5 km, 14.5–18.5 km and 18.5–22.5 km, for 30° N–90° N, including the latitude of the eruptions (~50° N). For liquid-phase $H_2SO_4$ four model results are presented for each volcanic eruption. The simulations differ concerning the implemented sedimentation (no sedimentation

and effective sedimentation radii of 0.1, 0.5, and 1 µm). In Fig. 6a–c, the sulfur mass is shown for $SO_2$ and sulfate aerosol separately, while Fig. 6d–f presents the total sulfur contained in $SO_2$ and sulfate aerosol. The total simulated sulfur mass is not influenced by chemical sulfur removal, but only by removal due to transport by advection and sedimentation. In Fig. 6, the eruption times of Kasatochi (7 Aug 2008) and Sarychev (12 Jun 2009) are indicated. Additionally, the eruption time of Redoubt (23 Mar 2009) is marked, as this eruption produces a signal in the measurements. It is not included in the simulations, however.

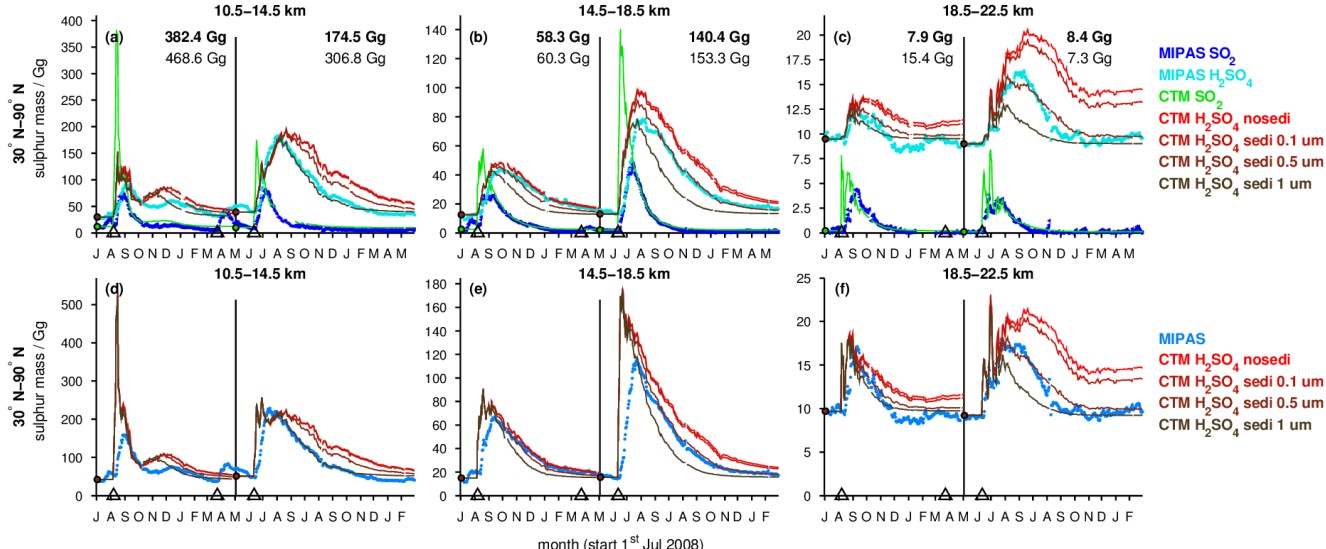

**Figure 6.** Sulfur mass contained in $SO_2$ and sulfate aerosol after the eruptions of Kasatochi (7 Aug 2008) and Sarychev (12 Jun 2009), from MIPAS measurements and CTM simulations. Presented are 5-days running means for 30° N to 90° N, in three altitude regions (10.5–14.5 km, 14.5–18.5 km, 18.5–22.5 km). For the CTM four simulations for $H_2SO_4$(liq) are shown with different effective sedimentation radii (0.1, 0.5, and 1 µm) and without sedimentation. All simulations are carried out with the volcanic $SO_2$ masses from Table 1 ("present study"). A constant background $SO_2$ and aerosol loading is derived from the measured data and added to the simulated sulfur masses per altitude region (indicated by circles). Model simulations for Kasatochi and Sarychev are separated by a vertical black line in May 2009. **(a) to (c)**: sulfur mass per species. **(d) to (f)**: sum of sulfur mass in $SO_2$ and sulfate aerosol. The numbers in (a) to (c) show the peak values of sulfur mass contained in simulated $SO_2$ on locations covered by MIPAS (bold), and for all available model data. Indicated by black triangles are the eruption days of Kasatochi (7 Aug 2008, 52.2° N/175.5° W), Redoubt (23 Mar 2009, 60.5° N/152.7° W), and Sarychev (12 Jun 2009, 48.1° N/153.2° E).

To analyse the measured and simulated data, data sets of sulfur mass densities (SMD = mass per unit volume) are resampled on a common grid with 1 km vertical spacing and a horizontal resolution that equals the model grid. On this new grid, the same data basis is used for the measured and simulated data, neglecting all "grid cells" for which either only MIPAS or only CTM data are available. For MIPAS aerosols, SMDs are calculated from the primarily retrieved volume densities, using an assumed aerosol density of 1,700 kg m$^{-3}$, and a binary solution of 75 wt% $H_2SO_4$–$H_2O$, while for MIPAS $SO_2$ and

the modelled values, SMDs are calculated from the mole-fractions. Sulfur masses are then derived from 5-days running zonal means of SMDs, by multiplication with the corresponding air volume of the new grid.

Generally, when calculating an integrated mass, high data coverage is crucial to prevent underestimation, therefore we use 5-days running zonal means. Zonal mean values, used to calculate sulfur masses, are derived using a method of increasing area averaging (see Appendix), to reduce the bias of mean values due to a non-uniform data coverage. Even though high data coverage is very important, we omit available data and information, as the same basis of available values is used for MIPAS and the CTM. This is appropriate when analysing the agreement between the data. Data are especially omitted for the CTM. Thus we also provide some information on modelled sulfur masses derived from the non-co-located data (Fig. 6a–c). The impact of missing data is strongest in the lowermost altitude region presented here. For MIPAS this is mainly due to the presence of clouds and ash, which were filtered out using the cloud filter by Spang et al. (2004) in the case of $SO_2$ and partly filtered out in the case of aerosol, and additionally the ice and ash filters by Griessbach et al. (2016 and 2014, respectively) for the aerosol retrieval. The CTM has low data coverage at lower altitudes due to its isentropic vertical grid. Interpolation to geometric heights starting at 10 km, produces missing values at altitudes up to 13 km.

To ease visual comparisons of measured and modelled sulfur mass in Fig. 6, a constant background is added to the model results, as only volcanic sulfur is considered in these simulations. The background mass is chosen considering the mass derived by MIPAS before the volcanic eruption in the region of interest, per altitude and latitude bin. This does not necessarily represent normal background conditions, but unmasks the anomalies caused by the volcanoes.

Concerning the measured and modelled $SO_2$ masses after the eruptions of Kasatochi and Sarychev (Fig. 6), comparisons show that until about one month after the eruptions, the $SO_2$ mass is by far underestimated by MIPAS. This underestimation of $SO_2$ was stressed by Höpfner et al. (2015), when comparing MIPAS $SO_2$ to measurements by the Microwave Limb Sounder (MLS), on board Aura (Pumphrey et al., 2015). It is mainly due to the presence of particles, that hinders MIPAS $SO_2$ measurements in largely eruption-affected air-parcels and causes a sampling bias towards less volcano-affected air parcels. Through our model simulations we confirm this bias, and the related time scale found by Höpfner et al. (2015). After this first month, the simulated $SO_2$ agrees well with the measurements by construction.

The measured decay of $SO_2$ is well reproduced by the CTM. Only oxidation by OH is considered in the model, and we see that the decay of $SO_2$ can adequately be described by this mechanism. Other processes, as decay by photolysis or reaction with atomic oxygen (O) are not considered, and following the good agreement between measurements and model results, can be neglected at the temporal and spatial scale of interest. Inside volcanic plumes, chemistry interactions might lead to changes in $SO_2$-lifetimes (Bekki, 1995). When a high amount of $SO_2$ gets depleted by hydroxyl radicals, the concentration of the radicals might decrease, which could reduce the speed of further depletion. The good accordance between MIPAS measurements and CTM simulations, which do not account for any feedback on the OH concentrations, suggests that even if such interactions occurred, they did not produce a strong impact in the timescale of months and larger spatial scales.

To investigate the effect of particle sedimentation on the residence time of sulfur after the volcanic eruptions, model simulations with different effective sedimentation radii are performed, as well as one simulation without any sedimentation. The radii lie in the range of aerosol size distributions as observed by Deshler et al. (2003) and Deshler (2008) for volcanically perturbed periods, and one constant radius is applied for all $H_2SO_4$ droplets per simulation. Fig. 6 shows the influence of varying the gravitational settling between no settling, and effective sedimentation radii of 0.1, 0.5, and 1 µm. The amount of sulfate aerosol removed by sedimentation increases with growing particle size, while the time needed for the removal increases for smaller effective sedimentation radii. The sulfur mass contained in liquid-phase $H_2SO_4$ from a simulation with an effective settling radius of 0.1 µm differs little from a simulation without sedimentation, while effective settling radii of 0.5, and 1 µm show an increasing impact. In the middle and uppermost altitude region, the best agreement between simulated and measured aerosols is found for an effective sedimentation radius of 0.5 µm, for both eruptions. At 10.5–14.5 km, especially in the case of Sarychev, the simulations show temporal disagreement to the decrease of measured aerosol. In the lowermost altitude range sparse data coverage has to be kept in mind, both for the measurements and model results. At these altitudes, sulfate aerosol simulated with a radius of 1 µm compares better. A larger effective sedimentation radius seems more appropriate at lower altitudes, as heavier particles can settle faster, and can be removed more rapidly than smaller and lighter particles. These can float in the atmosphere or undergo ascent. The particle size distributions of aerosols can further show natural variation for different volcanic eruptions; therefore some differences in the agreement between modelled and measured data when studying different volcanic eruptions can be expected. Model simulations show that compared to 10.5–18.5 km and compared to small particles, the bigger particles level out faster in the uppermost altitude range studied here. Reasons for this faster removal of the volcanic aerosol are that only little aerosol is injected in the altitude region 18.5–22.5 km, that bigger particles settle faster, and that settling velocities rise with increasing altitude due to the corresponding decrease in air density. In general, we conclude that an effective settling radius of 0.5 µm gives a satisfactory fit between the measurements and simulations for the purpose of studying sulfur mass and sulfur transport in the Northern hemisphere. Hence, we base all following model results on the CTM runs with an effective sedimentation radius of 0.5 µm.

We conclude from the comparisons between measured and simulated $SO_2$ and sulfate aerosol, that the amplitude of the peak of liquid-phase $H_2SO_4$ and its removal from the studied altitude regions, as measured by MIPAS, is consistent with the measured $SO_2$, both qualitatively and quantitatively. In the model, sulfur is released from $SO_2$ due to its reaction with OH, and sulfate aerosol is consequently formed. Modelled $SO_2$ that fits well to MIPAS $SO_2$ measurements in terms of amplitude and decay releases sulfur and builds $H_2SO_4$ that in turn matches well to MIPAS sulfate aerosol in terms of amplitude and decrease.

Further, we find that the dominating process on the evolution of volcanic sulfur is transport by the Brewer–Dobson circulation out of the region of interest. This becomes obvious when comparing the long-term removal of total modelled sulfur with and without sedimentation to the observed sulfur mass (Fig. 6d–f). In the case of the CTM, this excludes all influence by chemical reactions on the removal of volcanic sulfur. Even though consideration of sedimentation of sulfate aerosol with an effective sedimentation radius between about 0.5 and 1 µm further improves the agreement between model

results and observations in 10–22 km altitude, the decay of modelled sulfur mass without sedimentation already compares rather well with the measured decay of sulfur mass. Hereby we see that the removal is dominated by advection rather than sedimentation.

A peak can be seen in the measured and modelled sulfur dioxide and sulfuric acid masses in November / December 2008 (Fig. 6) in the lowermost altitude region (10.5–14.5 km). This peak is caused by downward transport of sulfur in the extra-tropics that has been emitted by the eruption of Kasatochi. In the following section (Sect. 4.2) more details are given on this transport pattern.

In the altitude region of interest, from around 10 to 22 km height, supplementary processes, as the photolysis of gas-phase $H_2SO_4$, that is important at altitudes above 30 km (Vaida et al., 2003; Brühl et al., 2015), or a meteoritic dust sink (Brühl et al., 2015), are not considered. Other processes, such as the evolution of sulfate aerosol through microphysical processes, as nucleation, coagulation, or condensation, and sedimentation of particles with different sizes can play a role in our region of interest. However, comparisons of simulations and measurements show that these processes are not essential to study the development of sulfur emitted by Kasatochi in 2008 and Sarychev in 2009.

## 4.2 Sulfur transport

The Kasatochi eruption injected a large amount of $SO_2$ directly into the stratospheric altitude region especially between 10 and 14 km (Table 1). Fig. 7 displays vertically resolved time series of $SO_2$ and liquid-phase $H_2SO_4$ mole-fractions for 30° N to 90° N, as measured by MIPAS (7a–c) and modelled by the CTM (7d–f), together with their sum (c and f). Both $SO_2$ and sulfate aerosol show a separation of the plume into a lower and upper part, in the measured and simulated data. While the lower part is removed relatively fast from the altitude range of observations, starting at 10 km, the upper part moves downward with time, following the Brewer–Dobson circulation. During the descent, the sulfur concentrations are reduced and some parts of the sulfur reach 10–12 km after several months. As the descent is seen in sulfate aerosol and $SO_2$, it is not primarily driven by sedimentation. A similar pattern of subsidence was found by Andersson et al. (2015, Fig. 3 therein) when studying aerosol scattering measured by CALIPSO (Cloud-Aerosol Lidar and Infrared Pathfinder Satellite Observations). Additionally shown in Fig. 7 is the eruption of Sarychev in 2009. No separation of the plume is noticeable, vertically the sulfur is distributed rather homogeneously. In terms of downward transport from higher altitudes in the extra-tropics and the removal from the stratosphere, $SO_2$ and sulfate aerosol from the Sarychev eruption evolve quite similar to the Kasatochi eruption.

Parts of the differences between the transport patterns after the eruptions arise from the injected $SO_2$ masses. In the case of Kasatochi, the main part of $SO_2$ was injected into altitudes below 14 km (518 Gg or 77 % of the injected $SO_2$). It is transported downwards and out of the region studied here relatively fast and therefore only a minor part is reflected in the aerosol loading. In the case of the Sarychev eruption, almost half of the $SO_2$ (367 Gg or 48 % of the injected $SO_2$) is injected into the altitude region above 14 km. It is available for conversion into sulfate aerosol for a longer period of time, as can be seen in the higher $H_2SO_4$ volume mixing ratios after the eruption of Sarychev, compared to Kasatochi. Model simulations

with "switched" SO$_2$ masses (mass of Kasatochi injected on the day and at the location of Sarychev, and vice versa), and a simulation with the SO$_2$ mass from the Kasatochi eruption injected at the location of Sarychev, reveal that the "double-plume" that has been observed after the eruption of Kasatochi results from the combination of the vertical distribution of injected SO$_2$ masses and the prevailing transport after the 7 Aug 2008, the eruption date of Kasatochi, in the model driven by
wind fields and heating rates. Neither of the simulations results in a comparable separation into an upper and lower part of the plume.

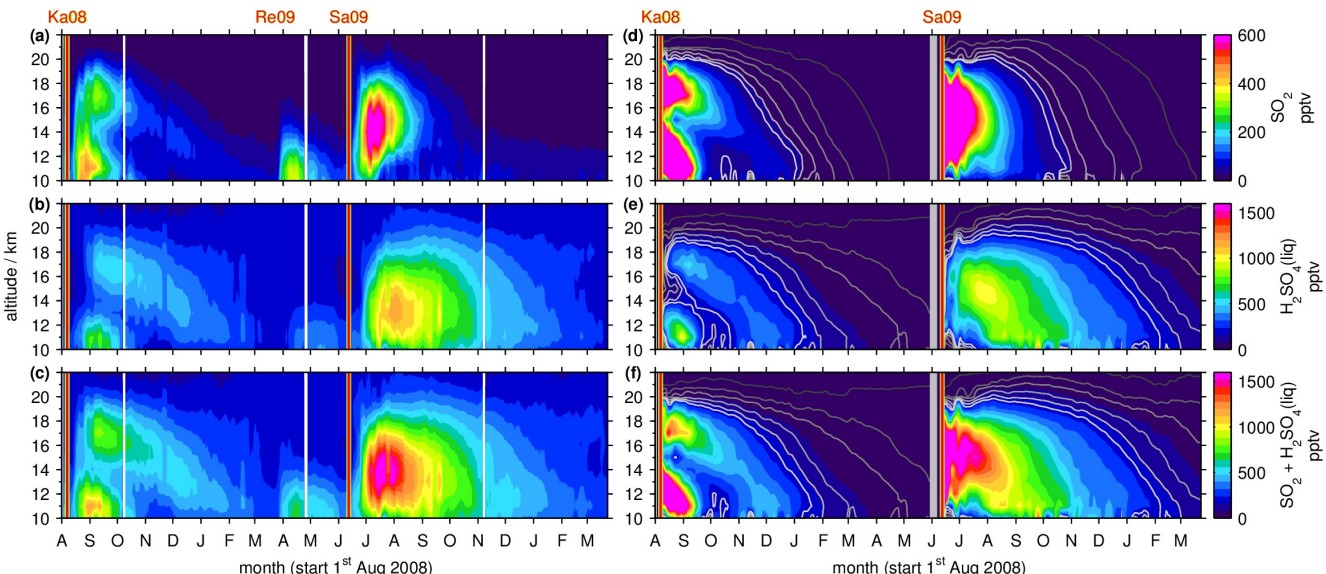

**Figure 7.** SO$_2$, liquid-phase H$_2$SO$_4$, and total sulfur (SO$_2$ + H$_2$SO$_4$) volume mixing ratios by MIPAS **(a–c)** and the CTM **(d–f)**. Vertically
resolved time series of SO$_2$ and sulfate aerosol for 30° N–90° N, starting on the 1 Aug 2008 (area weighted 5d running means). Indicated are the days of the eruptions of the volcanoes Kasatochi in Aug 2008, Redoubt in Mar 2009, and Sarychev in Jun 2009, which were observed by MIPAS. Redoubt is not considered in the model. The colour scale is restricted to 0–800 pptv (SO$_2$), and 0–1,600 pptv (sulfate aerosol, and sum of SO$_2$ and sulfate aerosol). Exceeding values are assigned with the respective limiting value. For the CTM contour lines in dark to light grey represent 1, 5, 10, 20, 30, and 40 pptv in the case of SO$_2$, and 5, 25, 50, 100, 150, and 200 pptv for sulfate aerosol, and
the sum of SO$_2$ and sulfate aerosol. For the CTM two simulations are shown, from August 2008 to May 2009, and from June 2009 to May 2010, separated by a bold grey line.

Figure 8 shows vertically resolved measurements and simulations of SO$_2$ and liquid-phase H$_2$SO$_4$ for 0° N–30° N. Some of the sulfur emitted at around 50° N reaches low latitudes. While the SO$_2$ is removed rather fast, compared to sulfate
aerosol, the sulfate aerosol resides in the tropics for many months and moves upwards with time. The modelled sulfate aerosol with an effective sedimentation radius of 0.5 µm behaves in a similar way as the measurements, moving slightly upwards with time. In comparison, a simulation with an effective sedimentation radius of 1 µm shows a rather horizontal

transport and faster removal, while the simulated lifting is stronger than in the measurements; when not considering sedimentation (not shown here). Due to uncertainties in modelled ascent speeds (e.g. Pommrich et al., 2010; Liu et al., 2013), the particle radius that is most suitable to reproduce MIPAS measurements by CTM simulations is not necessarily the best estimate when performing similar analyses with different models or meteorological driving data.

In the MIPAS data of the tropics, where the tropopause height is relatively constant at around 16–17 km, a clear transition from elevated sulfur mole-fractions in the troposphere to lower sulfur loading in the stratosphere is observed during times of weak volcanic influence (May–Jun 2009, Fig. 8c). The relatively high values at around 13–16 km in the measurements have already been noted in Fig. 5 and are supposed to only partly be connected to volcanic eruptions. A certain influence of elevated retrieved aerosol values due to cirrus clouds that have not been captured by the ice-filter (Sect. 3.1) is possible. To

which extent the observed enhancements in the measurements (Fig. 8a–c) are caused by the eruptions of Kasatochi and Sarychev is not clear. In the case of Kasatochi model simulations suggest that enhancements are confined primarily to altitudes above approximately 16 km. Additionally to the tropical enhancements at 13–16 km, the eruption of Dalafilla in November 2008 overlays with the observed sulfur that has been emitted by Kasatochi. The CTM simulations of Sarychev indicate that sulfur observed at altitudes as low as 12 km can be attributed to the volcanic eruption.

Differences between the presented zonally averaged measurements and model results arise partly from the fact that MIPAS measurements are not uniformly distributed and data were filtered, and due to sparse data coverage in the case of the CTM up to an altitude of 12–13 km. Data are partly missing in relatively large areas, which may lead to biased zonal means. In the measurements, for $SO_2$ data are missing particularly in the tropics at altitudes below about 15/16 km and at higher altitudes (up to ~17 km) in the region of the Asian Summer Monsoon. In the case of measured sulfate aerosol data are

filtered especially in the tropics at altitudes up to about 18/19 km and in the region of the Asian Summer Monsoon (up to ~20 km) and in polar regions entire profiles were filtered out due to PSCs. Especially after the eruption of Sarychev a higher sulfur content is simulated in the tropics compared to the measurements (Fig. 8) and enhancements are seen few days after the eruption. This results from a strong modelled meridional transport of $SO_2$ after this eruption. At about 12–16 km altitude the injected $SO_2$ reaches 15° N 7–8 d after the eruption. This strong southward transport early after the eruption is not

reflected in the measurements, which are, however, partly missing in the tropics due to filtering.

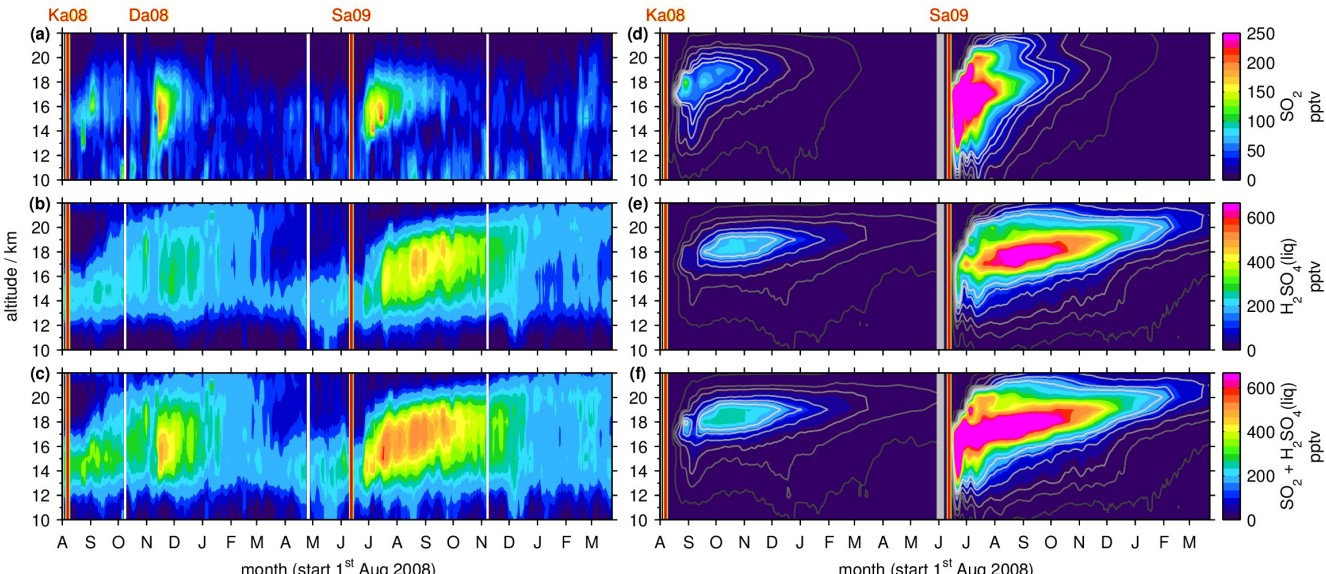

**Figure 8.** As Fig. 7, but for 0° N–30° N. The colour scales are restricted to 0–200 pptv (SO₂), and 0–800 pptv (sulfate aerosol, and sum of SO₂, and H₂SO₄(liq)). Additionally the volcano Dalafilla is indicated in Nov 2008, as it has been observed by MIPAS. It is not included in the model.

In Fig. 9 and 10 time series of latitudinally resolved mole-fractions show the transport of SO₂ and sulfate aerosol at different altitudes. We present time series of 5d running zonal mean mole-fractions for the Northern hemisphere, at altitudes from 10 to 22 km, both for MIPAS measurements and CTM simulations. As the model has low coverage at 10 km, results are not shown for the model at this altitude. For the eruption of Kasatochi, the separation of the plume and downward transport of the upper part is notable in mid- to high-latitudes, most easily visible for sulfate aerosol (Fig. 10).

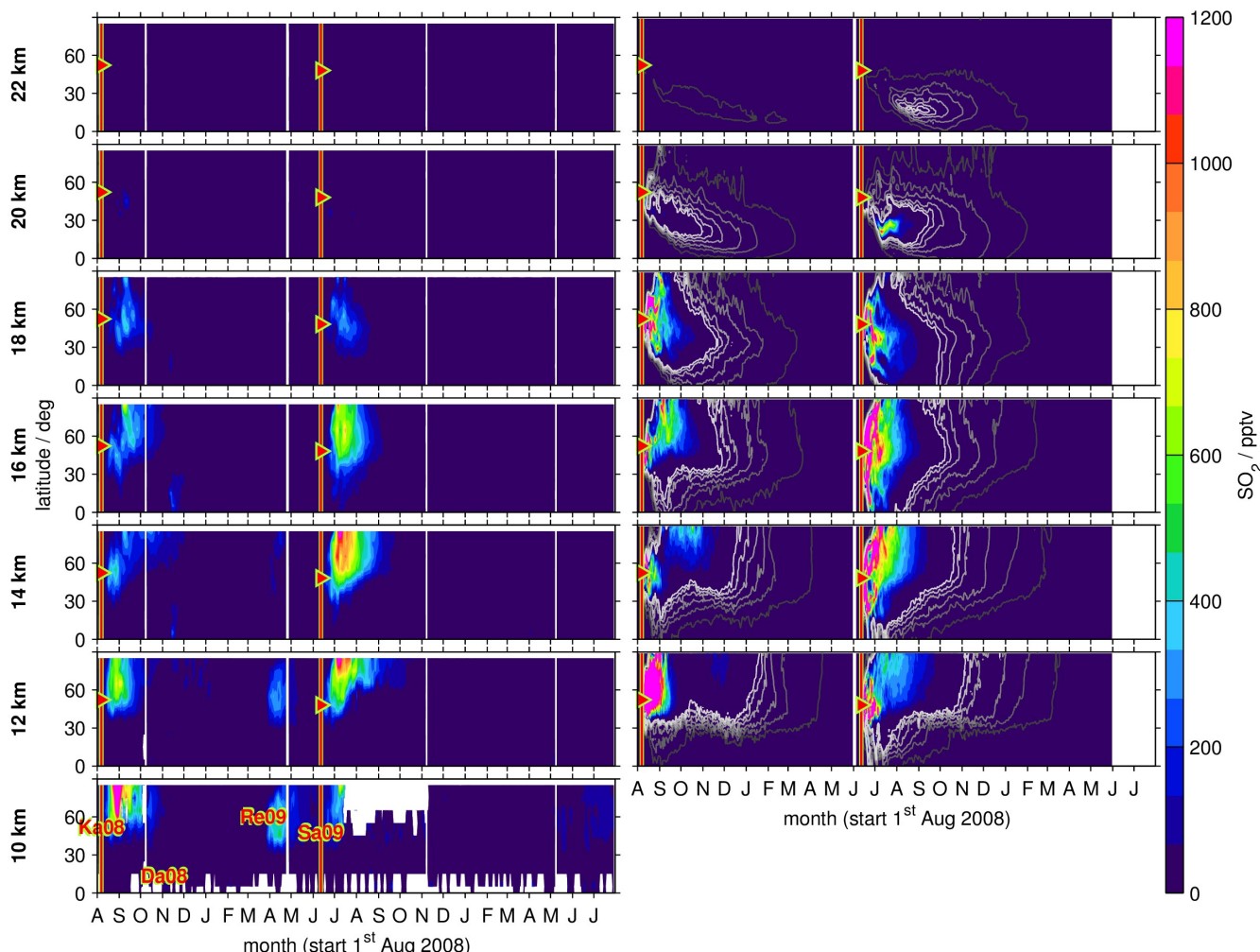

**Figure 9.** Time series of SO₂ from MIPAS measurements **(left panels)** and CTM simulations **(right panels)**. Shown are volume mixing ratios after the eruption of Kasatochi (7 Aug 2008) and Sarychev (12 Jun 2009), as zonal means for the Northern hemisphere, at various altitudes (10 to 22 km). For MIPAS 5d running means for 10° latitude bins are calculated, while for the CTM daily values are shown for ~2.5° latitude bins. The colour code is restricted to 0–1,200 pptv. Values exceeding these limits are assigned with the limiting value, respectively. Time and location of the eruptions of Kasatochi and Sarychev are indicated by red triangles. (left) Indicated are the eruptions of Kasatochi and Dalaffilla in 2008, and Redoubt and Sarychev in 2009, that were observed by MIPAS. (right) The two simulations are separated by a grey line. Black to white contour-lines denote 1, 5, 10, 20, 30, and 40 pptv. Only Kasatochi and Sarychev are included in the model.

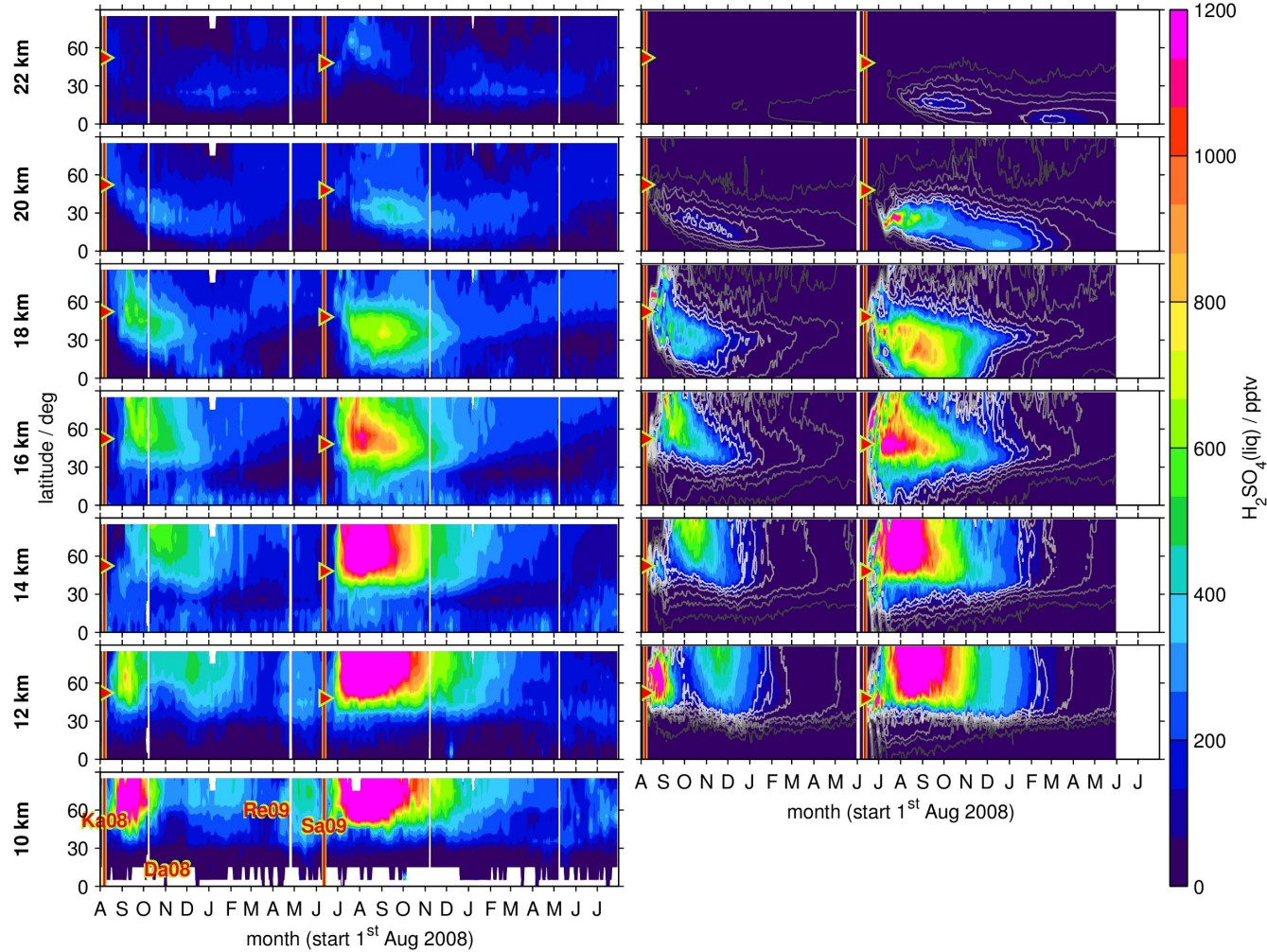

**Figure 10.** As Fig. 9, but for sulfate aerosol. (right) Black to white contour-lines denote 5, 25, 50, 100, 150, and 200 pptv.

Both in the measurements and simulations, most of the sulfur contained in $SO_2$ and sulfate aerosol stays north of 30° N at lower altitudes up to around 16 km (Fig. 9 and 10). Especially at low altitudes we find a mixing barrier at ~30° N, with a strong gradient between low values in the tropics and high values in the extra-tropics, which weakens towards higher

5    altitudes. This gradient is due to the subtropical jet stream, and is most easily detectable in the contour lines shown for modelled liquid-phase $H_2SO_4$ (Fig. 10, right panel), but similar patterns are observed by MIPAS. At around 16–18 km, especially in the longer lived sulfate aerosol, this forms a "tongue" of relatively high mole-fractions, which persists over a longer period than in the surrounding latitudes. At an altitude of about 18 km in the case of Kasatochi and ~16 km in the case of Sarychev, and at altitudes above, a southward transport of sulfur is observed (Fig. 9 and 10). At these altitudes, both the

10   MIPAS measurements and CTM simulations show that sulfur from mid-latitude volcanic eruptions can reach the tropics,

predominantly in the form of sulfate aerosol. In the tropics sulfur is then lifted in the "tropical pipe" (terminology following, e.g., Plumb 1996), and can reach the stratospheric "overworld" (terminology following, e.g., Hoskins, 1991), also seen in Fig. 8. Measurements of the stratospheric optical depth by the Optical Spectrograph and InfraRed Imager System (OSIRIS) onboard Odin also show that in the months after the eruptions of Kasatochi in 2008 and Sarychev in 2009 their impact extended to lower latitudes (Bourassa et al., 2012). Wu et al. (2017) studied the equatorward dispersion of the Sarychev volcanic plume together with the influence of the Asian summer monsoon on the transport pattern. They find that at 360–400 K potential temperature, the southward transport was primarily caused by anticyclonic Rossby wave breaking, intensified by the Asian summer monsoon during Northern hemisphere summer. Above 400 K less aerosol is transported into the tropics. They further show an "aerosol hole" in the anticyclone, surrounded by aerosol-rich air. Following Wu et al. (2017), a strong subtropical jet in combination with weak Rossby wave breaking events would hinder the southward transport of the volcanic plume during winter conditions. Compared to Sarychev, the southward transport of the Kasatochi eruption plume is weaker and initiates at higher altitudes. This might be explained by the eruption having been later during the monsoon season, leading to enhanced southward transport by the Asian summer monsoon for a shorter period of time. In model studies of the Sarychev eruption, Haywood et al. (2010) find that sulfate aerosol is transported around the entire globe in around 14 days. We see that the bulk of the aerosol that moves southwards reaches the equator about 2 to 3 months after the volcanic eruption. To some small extent this sulfur crosses the equator and can thereby influence the sulfur loading of the Southern hemisphere (see also Wu et al., 2017). Generally, similarities in the geographic pattern between Kasatochi in 2008 and Sarychev in 2009 have also been noticed by Haywood et al. (2010), caused by the agreement between the season, the injection altitude and latitude of the eruptions. The model results and MIPAS measurements of $SO_2$ and sulfate aerosol presented in our study confirm similarities in the transport patterns of sulfur after the volcanic eruptions of Kasatochi in August 2008 and Sarychev in June 2009, and a southward transport of the volcanic plumes, towards the equator, where sulfur can then ascent in the Brewer–Dobson circulation.

## 5 Discussion and conclusions

In this study a new data set of MIPAS/Envisat global aerosol volume densities and associated liquid-phase $H_2SO_4$ volume mixing ratios are presented for 2005 to 2012, covering the altitude range of 10 to 30 km, with up to 1,300 profiles per day. The MIPAS aerosol volume densities have been corrected for a positive bias in comparison to coincident balloon-borne in situ observations from Laramie, Wyoming. This bias is supposed to be caused by instrumental radiance baseline offsets. With absolute differences below $\pm 0.003 \ \mu m^3 \ cm^{-3}$ at 20–25 km, the bias corrected MIPAS profiles compare well with the in situ data. The strongest variability in the MIPAS sulfate aerosol is caused by various volcanic eruptions. Liquid-phase $H_2SO_4$ patterns from MIPAS are in general agreement with MIPAS $SO_2$ profiles from single limb-scans during volcanically perturbed and quiescent periods.

In a case study we investigate the evolution of volcanic sulfur after two major mid-latitude volcanic eruptions of the last decade (Kasatochi in 2008 at 51.2° N, and Sarychev in 2009 at 48.1° N) by combining this new data set with simultaneously observed profiles of $SO_2$ from the same instrument with the help of CTM simulations. Liquid-phase $H_2SO_4$ derived from the MIPAS aerosol retrieval is not only qualitatively, but also quantitatively consistent with the MIPAS $SO_2$ observed after the two volcanic eruptions. One of the advantages of deriving aqueous $H_2SO_4$ and $SO_2$ from one instrument is that sampling inconsistencies due to different geolocations and measurement times can be largely excluded. Some remaining sampling effects are caused by different filter methods, which depend partly on the retrieved species. The data sets provide a valuable basis for further analyses of the stratospheric sulfur loading. The new $H_2SO_4$ aerosol observations enable us to further constrain the total sulfur emitted into the stratosphere by the Kasatochi and Sarychev eruptions and to revise our previous estimates that were based on $SO_2$ observations only. The new estimates are 677 Gg $SO_2$ in the case of Kasatochi and 768 Gg $SO_2$ in the case of Sarychev that were injected into the altitude range 10–19 km. The decay of $SO_2$ after the volcanic eruptions and the formation of sulfate aerosol are consistent with known $SO_2$ chemical lifetimes due to reaction with OH, under OH background conditions (modelled OH climatology without any feedbacks between sulfur species and OH concentrations). While sedimentation of sulfate aerosol does play a role, the decay of sulfur in the mid-latitude lower stratosphere following the volcanic eruptions of Kasatochi and Sarychev is dominated by advective transport and transport by the Brewer–Dobson circulation. Sensitivity simulations with the CTM with different effective sedimentation radii indicate that the observed sulfate aerosol is best described by sedimentation of aerosol particles with an effective radius of about 0.5 µm. Most of the sulfur emitted by the two volcanic eruptions resides in the extra tropical lowermost stratosphere where it is transported downward across the tropopause. However, at higher altitudes (at about 16 to 22 km) parts of the volcanically emitted sulfur from these Northern hemisphere mid-latitude volcanoes is transported equatorwards where it is lifted in the "tropical pipe" into the stratospheric "overworld" and even enters the Southern hemisphere.

Our findings of the residence time and transport pathways of enhanced sulfate aerosol in the mid-latitude lower stratosphere have implications for the forcing of surface climate by moderate sized mid-latitude volcanoes and proposed geoengineering schemes. Sulfur injections into the lowermost stratosphere in mid-latitudes can affect not only the extra-tropics of the respective hemisphere, but are potentially transported towards the tropics, where they can undergo uplift and further transport by the Brewer–Dobson circulation, and can thereby reach the other hemisphere.

## 6 Data availability

The MIPAS data sets for aerosol volume densities and liquid-phase $H_2SO_4$ mole-fractions are available upon request from the authors or at http://www.imk-asf.kit.edu/english/308.php. Model results are available upon request from the authors.

## 7 Appendix

*Method of increasing areas for zonal averages:*

To reduce biasing of zonal averages due to non-uniformly distributed data, we use a method of increasing areas. It is based on the horizontal grid of our chemical transport model, which has a resolution of ~2.5° latitude x 3.75° longitude. For MIPAS daily arithmetic means are calculated for these ~2.5° x 3.75° boxes. These are then averaged to 32 boxes of ~ 10° x 11.25°. The area is further increased longitudinally by a factor of 2 in each step, while the number of boxes decreases by the same factor (32 x 11.25° → 16 x 22.5° → 8 x 45° → 4 x 90° → 2 x 180° → 1 x 360°). The result is not changed, compared to normal averaging (sum of values divided by number of values), when an equal number of values is available per ~ 2.5°x3.75° grid-cell, which is the case for the CTM, in altitudes above 13 km. By interpolating the model results to a vertical grid with 1 km resolution, starting at an altitude of 10 km, we find missing values up to 13 km. Therefore the method of increasing areas for zonal averages is applied to the simulated data in these altitudes as well. As the surface decreases with increasing latitude, the latitude bin is chosen to be 11.25° and not broader, as increasing latitude bins can give a too high weight to values corresponding to relatively smaller areas.

## 8 Author contributions

A. Günther developed and performed the model simulations, wrote most of the paper and conducted most of the analyses. M. Höpfner developed the MIPAS aerosol retrieval and provided the retrieval sensitivity studies and error estimations, and their description (Sect. 3.1). B.-M. Sinnhuber provided advice with the development and analysis of chemical transport modelling. T. von Clarmann, G. Stiller and M. Höpfner provided advice for the analyses of MIPAS data. S. Griessbach provided the flags for the MIPAS ice and ash filter. T. Deshler provided the balloon-borne in situ data and their description (Sect. 2.2). All authors contributed to the discussion of the results.

## 9 Competing interests

The authors declare that they have no conflict of interest. T. v. Clarmann and G. Stiller are ACP co-editors but have not been involved in the evaluation of this paper.

## 10 Acknowledgements

Parts of this work were supported by the European Commisions's Seventh Framework Programme (FP7/2007–2013) within the StratoClim project (grant no. 603557), National Science Foundation (award numbers 0437406 and 1011827), and the Helmholtz Association through the Programme Atmosphere and Climate (ATMO). Meteorological analysis data by

ECMWF and MIPAS level-1b calibrated spectra by ESA are acknowledged. The article processing charges for this open-access publication were covered by a Research Centre of the Helmholtz Association.

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
