# Peer review of "MIPAS observations of volcanic sulfate aerosol and sulfur dioxide in the stratosphere"

_Atmospheric Chemistry and Physics, 2017_

## Referee Comment (RC1) · H. C. Pumphrey (Referee) · 27 Jun 2017

**General remarks**

- The subject is an advance in knowledge, appropriate to the journal, and should be accepted, subject to minor corrections.

- The written English is clear and unambiguous, but has a rather stilted style and a sprinkling of grammatical errors. I note a couple of these below, but this is not a full proof-read.

- The figures are generally clear and well made; I have only a few suggestions for corrections.

[Figure]

**[ACPD](https://acpd)**

Interactive
comment

**Specific corrections**

I use the notation "P5L27" to mean page 5, line 27.

- All pages: It grieves me to point it out, as, to me, the -f- spelling of "sulfur" is a horrid Americanism which grates on the eye. But the journal's English guidelines state that *. . . it is our house standard to use the -f- spelling for sulfur (instead of sulphur) and related words for all varieties of English.*

- P3L16: The authors note that they use only the second of the two measurement periods, but do not spell out why. Was it not possible to estimate $SO_2$ from the first period data? Were there no volcanoes of interest during that period?

- P5L27 "The sulphur . . . builds $H_2SO_4$." The wording of this sentence and the use of the word "builds" in particular seems rather odd. A possible alternative wording is " The sulfur released from volcanic $SO_2$ reacts with OH to form $H_2SO_4$."

- P7, Figure 1: The vertical axis of the graph is not labelled and it is not clear to me whether it applies both to the refractive index curves and to the transmission curve.

- P9L16–18: I would remove the comma after "Both" and insert one after "increasing temperatures".

- Figure 3: The caption does not explain the difference between LPC 2m, LPC 1p and LPC 3m.

- Figure 5: It would be preferable to repeat the table of volcano names somewhere in this paper, rather than referring the reader to a different paper. Also, the levels in the filled contour plot are the rather odd choice of 100/7 units. The colour scale itself is a better choice than the dreadful "jet" scale that too many people still use.

[Figure]

But I feel that there might nevertheless be a better choice. In particular, I feel that it would be better for the colours at the upper end to become paler (e.g. red → magenta → almost-white) rather than tending towards a purple colour which is very close to the blue at the bottom of the scale. In making any such change it should be ensured that adjacent colours are clearly distinguishable from each other. (This is currently the case except, perhaps, for the shades of blue around 200 ppbv.)

- P13L2: "built" is rather an odd word choice. Maybe "produced" would be better.

- P15, table 1: Pumphrey's two estimates for Sarychev are the wrong way round, and one of them is missing its error. It should be $571 \pm 42$ above 147 hPa, and $1160 \pm 180$ above 215 hPa.

- P22L19: Remove comma after "Both".

- P24L11: "were" should be "where".

- P24L19: "hereby" should perhaps be "thereby"

---

## Author Comment (AC1) · 5 Jul 2017

We thank the reviewer Hugh Pumphrey for his helpful and constructive comments which we address in detail below.

The notation is as follows: P5L27 means page 5, line 27.

**General remarks**

- The subject is an advance in knowledge, appropriate to the journal, and should be accepted, subject to minor corrections.

[Figure]

- The written English is clear and unambiguous, but has a rather stilted style and a sprinkling of grammatical errors. I note a couple of these below, but this is not a full proof-read.

  Thank you for these corrections. We will try our best to improve the manuscript in this respect.

- The figures are generally clear and well made; I have only a few suggestions for corrections.

  We will take all those into consideration for the final version.

**Specific corrections**

- All pages: It grieves me to point it out, as, to me, the -f- spelling of "sulfur" is a horrid Americanism which grates on the eye. But the journal's English guidelines state that . . . it is our house standard to use the -f- spelling for sulfur (instead of sulphur) and related words for all varieties of English.

  Thank you very much for this reminder, we will use the "sulfur"-spelling in the revised version of the paper.

- P3L16: The authors note that they use only the second of the two measurement periods, but do not spell out why. Was it not possible to estimate $SO_2$ from the first period data? Were there no volcanoes of interest during that period?

  The $SO_2$ dataset by Höpfner et al. (2015) comprises retrieved sulphur dioxide profiles for both measurement periods. However, the first period is not considered within this study. We aimed at investigating two of the major mid-latitudinal eruptions (Kasatochi in 2008, and Sarychev in 2009) during the MIPAS measurement period from Jun 2002–Apr 2012. Volcanic eruptions during the first

period only injected SO$_2$ masses of below 100 Tg to 10–22 km (Höpfner et al., 2015). Furthermore, the much longer second measurement period (Jan 2005– Apr 2012) is characterised by a better vertical and horizontal resolution due to the denser vertical and horizontal limb sampling. Future work will be invested into the retrieval from the first period (Jun 2002–Mar 2004) in order to get an aerosol dataset covering the whole MIPAS lifetime.

The revised version of the paper will include the following sentences on P3L16: "Here we concentrate on the data from the second and longer measurement period (Jan 2005-Apr 2015), as the major mid-latitudinal volcanic eruptions between 2002-2012 occurred during this period. Furthermore, this measurement period is characterised by an improved vertical resolution, especially in the altitude region of the upper troposphere and lower stratosphere."

- P5L27 "The sulphur ... builds H$_2$SO$_4$" The wording of this sentence and the use of the word "builds" in particular seems rather odd. A possible alternative wording is "The sulfur released from volcanic SO$_2$ reacts with OH to form H$_2$SO$_4$."

This will be changed in the revised version.

- P7, Figure 1: The vertical axis of the graph is not labelled and it is not clear to me whether it applies both to the refractive index curves and to the transmission curve.

Thank you for making us aware of the missing label. The labels will be included in the revised version.

- P9L16–18: I would remove the comma after "Both" and insert one after "increasing temperatures".

This will be changed in the revised version.

- Figure 3: The caption does not explain the difference between LPC 2m, LPC 1p and LPC 3m.

In the caption of Fig. 3, we will clarify that the colour coding for the LPCs means that different Laser Particle Counters have been used for the measurements. "... measured by Laser Particle Counters (LPCs). Different LPCs have been used (colour-coded)."

- Figure 5: It would be preferable to repeat the table of volcano names somewhere in this paper, rather than referring the reader to a different paper. Also, the levels in the filled contour plot are the rather odd choice of 100/7 units. The colour scale itself is a better choice than the dreadful "jet" scale that too many people still use. But I feel that there might nevertheless be a better choice. In particular, I feel that it would be better for the colours at the upper end to become paler (e.g. red → magenta → almost-white) rather than tending towards a purple colour which is very close to the blue at the bottom of the scale. In making any such change it should be ensured that adjacent colours are clearly distinguishable from each other. (This is currently the case except, perhaps, for the shades of blue around 200 ppbv.)

A list of abbreviations is going to be added to the caption. Furthermore, the purple colours will be removed from the colour-scale in all contour plots, and we will consider updating the levels of the contour plots to match better to the values shown in the colour-bars.

- P13L2: "built" is rather an odd word choice. Maybe "produced" would be better.

This will be changed in the revised version.

- P15, table 1: Pumphrey's two estimates for Sarychev are the wrong way round, and one of them is missing its error. It should be 571±42 above 147 hPa, and 1160±180 above 215 hPa.

Thank you very much for this remark, will be updated in the revised version.

- P22L19: Remove comma after "Both".

This will be changed in the revised version.

- P24L11: "were" should be "where".

This will be changed in the revised version.

- P24L19: "hereby" should perhaps be "thereby"

This will be changed in the revised version.

---

## Referee Comment (RC2) · Anonymous Referee #2 · 11 Aug 2017

This study presents new measurements of aerosol volume densities and H2SO4 concentrations for 2005 to 2012 as obtained from MIPAS on-board ENVISAT. Using a chemical transport model (CTM), they also investigate the evolution of volcanic SO2 emitted from two volcanic eruptions in the northern mid-latitudes, eruptions of Kasatochi and Sarychev. This is a good paper that complements existing aerosol measurements and existing studies on investigating the volcanic eruption of Kasatochi and Sarychev. The paper presents new data sets that will be of interest to the readership of ACP. The paper would benefit from greater clarity in writing and from providing more information on the CTM model simulations, on the methodology of obtaining SO2 mass that has been used in the CTM simulations and on the bias correction that has been applied to the MIPAS data. After addressing my comments stated below, I recommend

[Figure]

the paper to be published in ACP.

General

The paper contains a number of spelling and grammatical errors. I only point out a few of them below and I would encourage the authors to re-work through the paper and correct all the errors. I would like to point out that the spelling of sulfur throughout the paper is incorrect. The journal guidelines clearly state: "In accordance with IUPAC, it is our house standard to use the -f- spelling for sulfur (instead of sulphur) and related words for all varieties of English."

I noticed that the authors use abbreviation/acronyms without defining them throughout the paper. I would encourage the authors to have a careful look through the paper and provide the definitions for the abbreviations used e.g. $H_2SO_4$ in the abstract and introduction. As stated in the journal guidelines, abbreviations ". . . need to be defined in the abstract and then again at the first instance in the rest of the text".

I know that it seems commonly accepted to write 'data is' but data is the plural of datum and therefore it should read 'data are'. Please check the wording throughout the paper. Also, 'dataset' should be corrected throughout the paper to 'data set'.

The abstract would benefit from a clearer structure. The authors describe the measurements briefly, go into the case study and then back again to the measurement. A clear 'story line' of what they did and what they have found is missing or it is not clear when the authors refer to the case study and then to measurements.

Specific comments

Page 1, ln 13: ". . . on board of the Environmental Satellite"

Remove 'of' and include (Envisat) at the end as I believe this satellite is mostly known by its acronym.

Page 1, ln 14/15: "The MIPAS aerosol dataset has been corrected for a possible

altitude-dependent bias by comparison with balloon-borne in situ aerosol measurements at Laramie, Wyoming."

I'm not sure about the word 'possible' in this sentence. Is there a bias or not and if there is why 'possible'? It is not clear from reading this sentence how the bias was corrected and what the Laramie measurement have to do with it? Was the bias discovered when comparing the measurements to the balloon measurements or was the comparison used to correct the bias, or both? This sentence needs to be made clearer. Please change the wording of 'possible altitude-dependent bias' throughout the paper.

Page 1, ln 15/16: 'The MIPAS data of stratospheric sulphate aerosol is linked to MIPAS observations of sulphur dioxide (SO2) with the help of Chemical Transport Model simulations.'

Replace 'is' with 'are'. Also, what do you mean by saying 'data are linked to MIPAS SO2 observations'? How can you link observations with CTM simulations?

Page 1, ln 16/17: 'We investigate the production of sulphate aerosol...'. '...and its fate from volcanically emitted SO2 for two volcanic case studies:'

Production of sulfate aerosol in the stratosphere I assume? For this you are using the CTM? To use 'its fate' is this sentence seems to be a rather odd word choice. Could the authors replace 'fate' throughout the paper?

Page 1, ln 20: 'While sedimentation of the sulphate aerosol plays a role, we find that the dominant mechanism controlling the stratospheric lifetime of sulphur after these volcanic eruptions at mid-latitudes is transport in the Brewer-Dobson circulation.'

This sentence needs to be reworded. How about: '... the lifetime of stratospheric sulfur is controlled mainly by the Brewer-Dobson circulation'.

Page 2, ln 5/6: 'Hofmann et al. (2009) observed an increase of stratospheric aerosol and speculated that this is due to anthropogenic emissions.'

Change 'increase of' to 'increase in'. Please also include 'increase in stratospheric aerosol load...' (or abundances). Use 'suggested' rather than 'speculated'.

Page 2, ln 6/7: 'Newer studies, however, show this increase to be connected more likely to a series of smaller and medium sized tropical volcanic eruptions (e.g. Neely et al., 2013).'

Reword to: '... show that this increase is likely to be connected to a number of small and medium sized volcanic eruptions located in the tropics.'

'Following Vernier et al. (2011), the increase of stratospheric aerosol levels since 2002 is connected to a series of moderate eruptions of volcanoes especially in the tropics.'

This seems repetitive and should be combined with the sentence before.

Page 2, ln 9: 'These volcanoes directly injected sulphur up to 20 km into the stratosphere'

Not clear what 'these' refers to and here a reference is needed that states that sulfur got injected into the stratosphere.

Page 2, ln 10: remove ',' after (2014).

'...noticed a strong contribution of aerosols in the lowermost stratosphere of the mid- and high latitudes to the volcanic aerosol forcing during the last decade'

It is not clear to me from this sentence what the authors are trying to say and what the message is.

Delete sentence: 'Understanding of stratospheric sulphur, its sources and sinks, and the processes involved in its conversion and transport is important in the framework of proposed climate engineering schemes (e.g. Niemeier and Timmreck, 2015; Rasch et al., 2008).'

Not sure why this is mentioned here and how that relates to this study.

Page 2, ln 14: The authors should define 'background conditions', i.e. that they mean 'non-volcanic' conditions.

Page 2 ln : Remove 'E.g.'

'Chin and Davis (1995), Thomason and Peter (2006), Bruhl et al. (2012),and Sheng et al. (2015), agree on a major contribution of OCS.'

Contribution to what?

'However, its exact contribution to stratospheric aerosol during background conditions is still in discussion.'

Replace with '. . . the magnitude to which OCS contributes to stratospheric aerosol loading . . .'

'During volcanically perturbed times volcanically emitted SO2 is the dominant source for stratospheric sulphate aerosol and causes most of the variability in the stratospheric sulphur content.'

The word 'volcanically' is used twice, why not just saying: "Volcanic eruptions are the dominant source for . . .". Replace 'content' with 'loading'. Are the authors referring here to the stratospheric sulfur concentrations or aerosol loading?

'In volcanic emissions, SO2 is the third most abundant emitted gas, after water vapour and carbon dioxide (von Glasow et al., 2009).'

Why is this relevant?

Page 2, ln 23: '. . . is useful'. Useful for what?

'From MIPAS several datasets that are relevant to the stratospheric sulphur content are already available.'

Replace 'content' with 'concentrations' or 'loading'. Replace 'datasets' with 'data sets' and please correct throughout the paper. Can the authors please clarify why MIPAS

measurements are relevant to the stratospheric sulfur loading? I believe the word choice here is misleading. Do the authors mean that MIPAS measurements are important to estimate the stratospheric sulfur loading?

Page 2: 'Here we present an additional dataset of sulphate aerosol from MIPAS, and combine the MIPAS SO2 and liquid-phase H2SO4 measurements with Chemical Transport Model (CTM) simulations to analyse the consistency of the two datasets, and the fate of volcanically emitted sulphur.'

Again, 'fate' is an odd word choice. It is not clear from this sentence what the CTM was used for and why. Are the authors here talking about two or three data sets?

Delete 'This paper has several purposes.'

Page 3, ln 3: Why did the authors choose the 2005-2012 period?

Page 3: 'We analyse MIPAS observations of SO2 and stratospheric sulphate aerosol in comparison to CTM simulations, and study the sulphur mass contained in SO2 and sulphate aerosol, together with the transport of their volcanic plumes.'

This sentence needs to be re-worded. It is not clear that this analysis is related to the volcanic eruptions stated in the sentence before.

'Finally, in Sect. 5 we draw final conclusions on the consistency between the MIPAS SO2 and the new MIPAS sulphate aerosol dataset,'

Does this only relate to the data sets during the volcanic eruptions? Either replace 'Finally' or 'final'.

Page 3, ln 23: It would be good if the authors could state the time period of the SO2 data set they are using in this study so that the reader doesn't need to look into Hoepfner et al. (2015).

Page 3, ln 28: Replace 'bias of' with 'bias between'

[Figure]

Page 3, ln 30: Replace 'within' with 'between'

'...when due to aerosol-related sampling artefacts the total mass of SO2 was found to be strongly underestimated (Hopfner et al., 2015).'

Re-word to: ' when the total mass of SO2 was found to be strongly underestimated due to aerosol-related sampling artefacts (..).'

'Their study comprises a dataset of volcanically emitted SO2 for 30 volcanic eruptions, as seen in the MIPAS measurements'

Not clear what 'their' refers to (I assume that is the study by Hoepfner et al.). Suggest to reword this sentence and also to write 'as observed by MIPAS' rather than 'seen in the MIPAS measurements'.

Should Section 3 be moved forward to 2.1.3? The structure of this paper is not clear to me. Section 2 was labelled 'Data sets and Methods' but then Section 3 is 'MIPAS aerosol data set'. That is confusing.

Page 4: 'The in situ measurements were completed with balloon-borne University of Wyoming optical aerosol counters and consist of size resolved aerosol concentrations from the surface to approximately 30 km.'

Replace 'completed' with 'made'... Suggest to change sentence to: "Size resolved aerosol concentration measurements from the surface to approx. 30 km altitude were made with ..."

Page 4: Reword sentence 'The latest style of the three was used initially in 2006, became the standard Laramie instrument in 2008, and was flown on quasi-Lagrangian balloons in Antarctica in 2010 (Ward et al., 2014).'

Something is missing in this sentence.

Page 4, ln 12/13: 'For the MIPAS validation the Wyoming measurements were confined to those made with this final instrument (Laser based counter, LPC), which measures

particles with radii > 0.08–4.2 $\mu$m in eight size classes."

Why were the data from this LPC used? Why not any measurements from other OPC measurements before 2006?

Page 4, ln 15 and following: 'To derive geophysical quantities from the size resolved aerosol concentration measurements it requires fitting a size distribution to the data. In the past this has been done by choosing a subset of the measurements to fit either a unimodal or bimodal lognormal size distribution. The final size distribution selected is from that subset of the measurements which minimizes the root mean square error when the fitted distribution is compared to all the measurements. This approach has recently been changed to use laboratory measurements of the counting efficiency at each channel and then search the lognormal parameter space for the lognormal coefficients, which minimizes the error of the fitted distribution compared to the measurements.'

How does the 'new' approach compare to the 'old' approach? How different are the measurements when derived with the new method compared to the old approach? And why is the new approach not used but mentioned here?

Page 4, ln 24: 'The isentropic Chemical Transport Model used in our study...'

Delete 'isentropic'. It is a CTM using isentropic levels as vertical coordinates but does that mean it could not be used differently? Include CTM acronym.

Page 4, ln 30: What are the initial OCS concentrations and where were they taken from? It would help the reader if the chemical reactions included in the CTM would be listed somewhere. Does the model include liquid and gas-phase chemistry? Are you only considering chemistry in the stratosphere? You state 10 to 55 km, and depending on where you are on the globe, 10 km could still be in the upper troposphere. So does cloud uptake of SO2 play a role here? If not, why not?

Page 5, ln 20: Do the authors believe that it will be clear to the reader what MI-

[Figure]

PAS/Balloon is in this context, especially since it is not mentioned before? I would suggest to add more information on this here and also to refer to it at MIPAS-B.

Page 6, ln 2: Remove 'E.g.'

Figure 3: Is it possible to show the uncertainties on the in situ measurements? What does LCP 2m, 1p and 3m stand for? The standard deviations on the mean values are shown but that is not the uncertainty on the measurement which would be interesting to have a look at? Wouldn't it be more appropriate to calculate the uncertainty on the mean using the measurement uncertainties on each datum that went into the mean calculation?

Page 10, ln 3: 'In Fig. 4a the standard errors of the mean show the uncertainty of the bias.'

I don't understand this statement. How does a standard error of the mean profile relates to the uncertainty of the bias? This needs to be clarified.

Figure 3 shows a strong signal at around 18 km in the balloon measurements on 28.07.2011 which does not seem to reflect in any way in the mean and its standard error shown in Figure 4a. Is that expected? This takes me back to the point that the mean values and their standard error do not take into account the uncertainty on each measurement.

What did the authors do to de-bias MIPAS measurements shown in Figure 3 and 4? There are some words around it on Page 10 (ln 5 to 10) but this didn't answer the question about what the authors actually did.

Page 13, ln 1: 'Sulphur is released from OCS mainly in the tropics at altitudes between about 25 and 35 km (Bruhl et al., 2012) and the sulphate aerosol that is built is transported towards mid-latitudes and lower altitudes.'

Is that statement still true with new publications (e.g. Lennartz et al. (2017)) about OCS being published?

Page 13: 'In the mid-latitudes of the Northern Hemisphere, the sulphate aerosol is increased during boreal summer at around 10–12 km.'

I'm not sure if that is a strong enough signal as the Northern Hemisphere is strongly disturbed by volcanic eruption according to Figure 5. What could be a cause for this increase?

Figure 5: There is a strong signal at altitudes 18 to 22 km (around 2007), i.e. enhanced H2SO4 concentrations which cannot be seen in the SO2 concentrations. Do the authors have an explanation for this signal in the tropics that high up?

Page 14, ln 8: Is 'sedimentation radius' the right term to be used? I don't think it is (I think the authors mean the radius of the aerosol) and the authors might want to think about rewording this sentence and where appropriate throughout the paper.

Page 14, ln 9: 'Good accordance between the modelled and measured SO2 masses is essential to test, . . .'

This sentence is rather odd and 'accordance' should be replaced with 'agreement'?

Page 14, ln 11: 'In Table 1, SO2 masses for three altitude regions. . .' Include 'injected SO2 amounts for three altitude regions as used in the CTM simulations. . .' The authors do not explain why and how they have chosen these amounts. Please clarify.

Page 14, ln 12: 'The simulations result in good agreement between measured and modelled SO2.'

Can the authors please clarify what simulations they are talking about? What is the simulation period? What the time step? I feel that there are more information required about the CTM simulations.

Page 14, ln 14: 'This method was applied as in the first month after the eruption MIPAS underestimates the SO2 (Hopfner et al., 2015).'

I don't understand what the authors are trying to say here. This method was applied

in their study or the study of Hoepfner et al., (2015). Why is it important to know how Hoepfner et al. derived the mass? How did the authors derive the SO2 mass used for their study? Could the authors please clarify why they didn't use the same values as Hoepfner et al. and how they derived their values for SO2 mass injected by the two volcanoes? And why do they not provide any uncertainties on their values?

Figure 6: Wouldn't a sedimentation radius of 0 mean that there are no aerosols? Do the authors mean that aerosol doesn't get lost through sedimentation? I don't understand why the eruption of Redoubt (23 Mar 2009, 60.5° N/152.7° W) is included here but was never mentioned before? Also, it is not clear if additional SO2 has been injected in the model due to this eruption (I suspect not). If the authors want to include Redoubt in the figure because there is a signal in the measurements, they should state that clearly in the text.

What in the CTM causes SO2 to be lost so much faster between 10.5 and 14.5 km compared to 14.5 and 18.5 km? Again, I think more information about the CTM is required. Can the authors explain why for a radius of $1\mu$m the H2SO4 loss is leveling off earlier (around November) between 18.5 and 22.5 km than compared to the other altitude ranges? What is causing the second peak (around Dec) in H2SO4 between 10.5 and 14.5 km? This peak is also seen in the MIPAS measurement but shifted by half a month. The peak in SO2 as modelled by the CTM is shifted compared to the peak in MIPAS SO2. Can the authors explain why that is?

Page 18, ln 15: '...due to its reaction with OH, and sulphate aerosol is consequently built.'

This sentence needs to be reworded – use formed rather than build.

Page 18, ln 23/24: '...the modelled sulphur mass without sedimentation already compares rather well with the measured sulphur mass.'

Looking at Figure 6d to f, are the authors really saying that the no-sedimentation run

compares well to the measurements, especially in 18.5 to 22.5 km region? I do not agree with this statement and their conclusions drawn from this. The fit to the measurements is better for the runs where sedimentation was considered.

Figure 7: Again why is the Redoubt eruption included in this Figure. This is the first time where the authors mention the simulation periods. This has to come earlier and it has to be described in the text. Why is the pattern for H2SO4 (CTM simulation) for the Kasatochi eruption so different from the Sarychev eruption? For the Kasatochi eruption the H2SO4 concentrations are lower than for the Sarychev eruptions although similar amounts of SO2 where injected? How much of the simulated Kasatochi 'double-plume' is made due to the choice of the SO2 mass being injected at different altitude levels? The injection heights of SO2 for the Sarychev eruption seem not appropriate when compared with the measurements. The model simulations show high SO2 concentrations from 10 to 19 km while in the observations the range goes from 11 to 16 km. Did the authors conduct any sensitivity studies regarding the injection height and the impact on their results?

Page 19: "and reaches 10 km after a few months."

The upper plume doesn't reach 10 km after a few months as shown in Figure 7. It is moving down but by doing so that sulfur concentration reduces. Or are the authors talking about the model simulations?

Figure 8: The CTM model shows about 1 month lag before SO2/H2SO4 is seen in the tropics compared to the observations which show a signal right from the beginning although the signal in the observations might not be from the volcanoes investigated here? The signal in the measurements in August must be from a different volcano? For the Sarychev eruption, the model seems to show the signal in the tropics earlier than it is seen in the measurements. Why could that be? The lag (time lag for when the eruptions is seen in the tropics) seems to be reduced for the Sarychev eruption? The model seems to overestimate the peak amount of SO2+H2SO4 compared to the

observations. Any explanation for that given that the model agrees with the peak values of measurements better between 30 and 90N?

Figure 9: For the CTM simulations, it looks like SO2 is immediately enhanced at lower latitudes (down to 15deg N) at the time of the eruption, especially for the Sarychev eruption. Why is that? SO2 is much more confined to the NH in the observations. How is the SO2 emitted by the volcanoes injected into the model? The authors state that 'SO2 is uniformly distributed to the grid boxes per altitude range' but then I don't understand why the picture for Sarychev at 12 and 14 km is so different from the picture for Kasatochi given that SO2 was uniformly distributed between 10 and 14 km (just looking at the time of the eruption, realizing that things will change after some time)? I find the differences in the latitudinal distribution of SO2 between the observations and the model interesting and would like the authors to comment on that.

Page 22, ln20: 'Especially at low altitudes we find a mixing barrier at $\sim$30° N, with a strong gradient between low values in the tropics and high values in the extra-tropics, which weakens towards higher altitudes.'

Which seems to be more pronounced in the measurements than in the CTM, especially for SO2. Do the authors have any explanation for the 'leak' towards the equator in the model, especially in July/Aug after the Sarychev eruption?

Page 22, ln 25: 'An additional transport process starts at an altitude of about 18 km in the case of Kasatochi and $\sim$16 km in the case of Sarychev (Fig. 9 and 10).'

What 'additional' transport process are the authors referring to here?

Figure 10: Enhanced H2SO4 concentrations reach further South in the model simulations than the observations show (at 18 km). In the observations H2SO4 is more confined to latitudes between 15 and 60deg N while for the model simulations, enhanced H2SO4 concentrations reach the equator. Why the difference?

Page 23, ln 12: 'The weaker southward transport in the case of the Kasatochi eruption

that starts at higher altitudes, compared to the Sarychev eruption, could be due to the eruption having been later during the monsoon season, leading to enhanced southward transport by the Asian summer monsoon for a shorter period of time.'

This sentence is confusing... 'weaker transport leading to enhanced transport'? Could the authors please clarify this sentence. I think I know what the authors mean but they need to clarify which eruption they are talking about in the second part of the sentence 'could be due to the eruption'.

Page 23, ln 20: 'In this study a new dataset of MIPAS/Envisat global aerosol volume densities and liquid-phase H2SO4 VMR distributions is presented for 2005 to 2012...'

Are the authors not talking about two data sets that are presented? Replace 'is' with are and use data sets instead of dataset.

Page 24: 'The new H2SO4 aerosol observations enable us to further constrain the total sulphur emitted into the stratosphere by the Kasatochi and Sarychev eruptions and to revise our previous estimates that were based on SO2 observations only.'

So what is the new estimate of total sulfur emitted into the stratosphere by both volcanoes? It would make sense to include that in the conclusions.

Page 24, ln 6: '...under OH background conditions'.

What do the authors refer to when saying 'OH background conditions'? That is not clear to me.

---

## Author Comment (AC2) · 16 Sep 2017

We thank Referee #2 for the detailed and constructive comments (in black). We address the suggested improvements in detail below (in blue).

This study presents new measurements of aerosol volume densities and $H_2SO_4$ concentrations for 2005 to 2012 as obtained from MIPAS on-board ENVISAT. Using a chemical transport model (CTM), they also investigate the evolution of volcanic $SO_2$ emitted from two volcanic eruptions in the northern mid-latitudes, eruptions of Kasatochi and Sarychev. This is a good paper that complements existing aerosol measurements and existing studies on investigating the volcanic eruption of Kasatochi

[Figure]

and Sarychev. The paper presents new data sets that will be of interest to the readership of ACP. The paper would benefit from greater clarity in writing and from providing more information on the CTM model simulations, on the methodology of obtaining $SO_2$ mass that has been used in the CTM simulations and on the bias correction that has been applied to the MIPAS data. After addressing my comments stated below, I recommend the paper to be published in ACP.

**General**

- The paper contains a number of spelling and grammatical errors. I only point out a few of them below and I would encourage the authors to re-work through the paper and correct all the errors.

  Thank you for the error–corrections given in the following. We will take all those into consideration for the revised version, and will try our best to improve the manuscript in this respect.

  I would like to point out that the spelling of sulfur throughout the paper is incorrect. The journal guidelines clearly state: "In accordance with IUPAC, it is our house standard to use the -f- spelling for sulfur (instead of sulphur) and related words for all varieties of English."

  Thank you very much for this reminder, we will use the "sulfur"-spelling in the revised version of the paper.

- I noticed that the authors use abbreviation/acronyms without defining them throughout the paper. I would encourage the authors to have a careful look through the paper and provide the definitions for the abbreviations used e.g. $H_2SO_4$ in the abstract and introduction. As stated in the journal guidelines, abbreviations "... need to be defined in the abstract and then again at the first instance in the rest of the text".

The updated version of the manuscript will contain the definitions for abbreviations, both in the abstract and main text.

- I know that it seems commonly accepted to write 'data is' but data is the plural of datum and therefore it should read 'data are'. Please check the wording throughout the paper. Also, 'dataset' should be corrected throughout the paper to 'data set'.

  This will be corrected throughout the paper.

- The abstract would benefit from a clearer structure. The authors describe the measurements briefly, go into the case study and then back again to the measurement. A clear 'story line' of what they did and what they have found is missing or it is not clear when the authors refer to the case study and then to measurements.

  Thank you for this advice.

  We will change the structure of the abstract as follows: on page 1, ln 15–16 ("The MIPAS data of stratospheric sulfate aerosol are linked to MIPAS observations of sulfur dioxide ($SO_2$ with the help of Chemical Transport Model (CTM) simulations.") will be deleted. Instead, on page 1, ln 19 "... during boreal summer. 'With the help of Chemical Transport Model (CTM) simulations for the two volcanic eruptions, we' show that ..." will be added.

**Specific comments**

- Page 1, ln 13: "... on board of the Environmental Satellite"

  Remove 'of' and include (Envisat) at the end as I believe this satellite is mostly known by its acronym.

  This will be changed in the revised version.

- Page 1, ln 14–15: "The MIPAS aerosol dataset has been corrected for a possible altitude-dependent bias by comparison with balloon-borne in situ aerosol measurements at Laramie, Wyoming."

I'm not sure about the word 'possible' in this sentence. Is there a bias or not and if there is why 'possible'? It is not clear from reading this sentence how the bias was corrected and what the Laramie measurements have to do with it? Was the bias discovered when comparing the measurements to the balloon measurements or was the comparison used to correct the bias, or both? This sentence needs to be made clearer. Please change the wording of 'possible altitude-dependent bias' throughout the paper.

MIPAS aerosol data show a bias in comparison to the in situ data. The de-biasing is made based on this comparison. The word 'possible' will be deleted in the revised version (throughout the paper).

The information on page 1, ln 14–15 will be changed to: "In comparison to balloon-borne in situ measurements of aerosol at Laramie, Wyoming, the MIPAS aerosol data have a positive bias that has been corrected, based on the difference to the in situ data." On page 23, ln 23 we will replace "The MIPAS aerosol volume densities have been corrected for possible instrumental radiance baseline offsets by comparison to coincident balloon-borne in situ observations from Laramie, Wyoming." by "The MIPAS aerosol volume densities have been corrected for a positive bias in comparison to coincident balloon-borne in situ observations from Laramie, Wyoming. This bias is supposed to be caused by instrumental radiance baseline offsets."

- Page 1, ln 15–16: 'The MIPAS data of stratospheric sulphate aerosol is linked to MIPAS observations of sulphur dioxide ($SO_2$) with the help of Chemical Transport Model simulations.'

Replace 'is' with 'are'. Also, what do you mean by saying 'data are linked to

MIPAS SO$_2$ observations'? How can you link observations with CTM simulations?

'is' will be replaced by 'are'.

On page 1, ln 15–16 "The MIPAS data of stratospheric sulfate aerosol are linked to MIPAS observations ... simulations." will be deleted. Instead, "'With the help of Chemical Transport Model (CTM) simulations of the two volcanic eruptions, we' show that the MIPAS sulfate aerosol and SO$_2$ ... ." will be added on page 1, ln 19.

- Page 1, ln 16–17: 'We investigate the production of sulphate aerosol ...'. '... and its fate from volcanically emitted SO$_2$ for two volcanic case studies:'

Production of sulfate aerosol in the stratosphere I assume? For this you are using the CTM? To use 'its fate' in this sentence seems to be a rather odd word choice. Could the authors replace 'fate' throughout the paper?

We study the sulfate aerosol predominantly in the stratosphere, right, and the CTM and MIPAS data are used in the case study on the two volcanic eruptions to do so.

For clarification we will change the manuscript as follows: "... 'stratospheric' sulfate aerosol ..." will be added to the sentence. Page 1, ln 15–16 "The MIPAS data of stratospheric sulfate aerosol are linked to MIPAS observations ... simulations." will be deleted. Instead, "'With the help of Chemical Transport Model (CTM) simulations of the two volcanic eruptions, we' show that the MIPAS sulfate aerosol and SO$_2$ ... ." will be added on page 1, ln 19.

We will replace the word 'fate' throughout the paper. In this special case no replacement will be done, but "and its fate" will be deleted. ("We investigate the production of sulfate aerosol from volcanically emitted SO$_2$ ...") Depending on the sentence, 'fate' is replaced by 'evolution' or 'development'.

- Page 1, ln 20: 'While sedimentation of the sulphate aerosol plays a role, we find that the dominant mechanism controlling the stratospheric lifetime of sulphur

after these volcanic eruptions at mid-latitudes is transport in the Brewer-Dobson circulation.'

This sentence needs to be reworded. How about: '... the lifetime of stratospheric sulfur is mainly controlled by the Brewer-Dobson circulation'.

This will be changed in the revised version.

("While sedimentation of the sulfate aerosol plays a role, we find that the long-term decay of stratospheric sulfur after these volcanic eruptions at mid-latitudes is controlled mainly by transport in the Brewer-Dobson circulation.")

- Page 2, ln 5–6: 'Hofmann et al. (2009) observed an increase of stratospheric aerosol and speculated that this is due to anthropogenic emissions.'

Change 'increase of' to 'increase in'. Please also include 'increase in stratospheric aerosol load ...' (or abundances). Use 'suggested' rather than 'speculated'.

This will be changed in the revised version.

("Hofmann et al. (2009) observed an increase in stratospheric aerosol load and suggested that this is due to anthropogenic emissions.")

- Page 2, ln 6–7: 'Newer studies, however, show this increase to be connected more likely to a series of smaller and medium sized tropical volcanic eruptions (e.g. Neely et al., 2013).'

Reword to: '... show that this increase is likely to be connected to a number of small and medium sized volcanic eruptions located in the tropics.'

'Following Vernier et al. (2011), the increase of stratospheric aerosol levels since 2002 is connected to a series of moderate eruptions of volcanoes especially in the tropics.'

This seems repetitive and should be combined with the sentence before.

This will be changed to: "Newer studies, however, show that this increase is likely to be connected to a number of small and medium sized volcanic eruptions especially in the tropics (e.g. Neely et al., 2013; Vernier et al., 2011)"

- Page 2, ln 9: 'These volcanoes directly injected sulphur up to 20 km into the stratosphere'

  Not clear what 'these' refers to and here a reference is needed that states that sulfur got injected into the stratosphere.

  This will be changed to: "During the last decade several volcanoes directly injected sulfur up to 20 km into the stratosphere (Vernier et al., 2011)."

- Page 2, ln 10: remove ',' after (2014).

  The comma will be removed in the revised version.

- '... noticed a strong contribution of aerosols in the lowermost stratosphere of the mid- and high latitudes to the volcanic aerosol forcing during the last decade'

  It is not clear to me from this sentence what the authors are trying to say and what the message is.

  To clarify what we intended to say the sentence will be replaced by: "Ridley et al. (2014) and Andersson et al. (2015) emphasise the importance of volcanic aerosol in the lowermost stratosphere at mid- and high-latitudes on the total volcanic aerosol forcing during the last decade. Their studies show that stratospheric altitudes below ∼15 km (380 K isentrope), which are not represented in many of the aerosol data sets, need to be taken into consideration when studying the global radiative forcing generated by volcanic eruptions in the extra-tropics."

- Delete sentence: 'Understanding of stratospheric sulphur, its sources and sinks, and the processes involved in its conversion and transport is important in the

framework of proposed climate engineering schemes (e.g. Niemeier and Timm-reck, 2015; Rasch et al., 2008).'

Not sure why this is mentioned here and how that relates to this study.

This information might be misplaced here and will therefore be deleted.

Instead on page 1, ln 2 we will add the following: "... for climate change modelling studies. 'Increased interest in stratospheric sulfate aerosol is also connected to its potential use in climate engineering schemes (e.g. Niemeier and Timmreck, 2015; Rasch et al., 2008).' " as we think that a reference to climate engineering studies should be contained in this paper.

- Page 2, ln 14: The authors should define 'background conditions', i.e. that they mean 'non-volcanic' conditions.

In the revised version we will add this information "background / non-volcanic conditions".

- Page 2 ln : Remove 'E.g.'

This will be removed in the revised version.

- 'Chin and Davis (1995), Thomason and Peter (2006), Brühl et al. (2012),and Sheng et al. (2015), agree on a major contribution of OCS.'

Contribution to what?

We refer to the contribution of OCS to stratospheric sulfate aerosol.

"contribution of OCS 'to stratospheric sulfate aerosol'." will be added in the revised version.

- 'However, its exact contribution to stratospheric aerosol during background conditions is still in discussion.'

Replace with '... the magnitude to which OCS contributes to the stratospheric aerosol loading ...'

This will be changed in the revised version.

- 'During volcanically perturbed times volcanically emitted $SO_2$ is the dominant source for stratospheric sulphate aerosol and causes most of the variability in the stratospheric sulphur content.'

The word 'volcanically' is used twice, why not just say: "Volcanic eruptions are the dominant source for ...". Replace 'content' with 'loading'. Are the authors referring here to the stratospheric sulfur concentrations or aerosol loading?

We are referring to both, the stratospheric sulfur and aerosol loading.

The sentence will be changed to: "By emitting $SO_2$, volcanic eruptions are the dominant source for stratospheric $SO_2$ (direct) and sufate aerosol (indirect) under non-background conditions, and cause most of the variability in the stratospheric sulfur loading."

- 'In volcanic emissions, $SO_2$ is the third most abundant emitted gas, after water vapour and carbon dioxide (von Glasow et al., 2009).'

Why is this relevant?

This additional information on volcanic emissions will be deleted in the revised version.

- Page 2, ln 23: '... is useful'. Useful for what?

A combination of observations and model simulations is useful for studies of stratospheric sulfur. However, we decided on deleting this sentence in the revised version.

- 'From MIPAS several datasets that are relevant to the stratospheric sulphur content are already available.'

Replace 'content' with 'concentrations' or 'loading'. Replace 'datasets' with 'data sets' and please correct throughout the paper.

This will be changed in the revised version.

Can the authors please clarify why MIPAS measurements are relevant to the stratospheric sulfur loading? I believe the word choice here is misleading. Do the authors mean that MIPAS measurements are important to estimate the stratospheric sulfur loading?

You are right, the word choice is misleading.

"From MIPAS several data sets 'of trace gas species' that are relevant to 'study' the stratospheric sulfur loading are already available." will be added.

- Page 2: 'Here we present an additional dataset of sulphate aerosol from MIPAS, and combine the MIPAS $SO_2$ and liquid-phase $H_2SO_4$ measurements with Chemical Transport Model (CTM) simulations to analyse the consistency of the two datasets, and the fate of volcanically emitted sulphur.'

Again, 'fate' is an odd word choice. It is not clear from this sentence what the CTM was used for and why. Are the authors here talking about two or three data sets?

Basically we are talking about two data sets here. The MIPAS $SO_2$ and $H_2SO_4$ data sets. MIPAS $SO_2$ is retrieved as volume mixing ratios and the aerosol data set consists of aerosol volume densities (also converted into VMR). These are compared to each other (2005–2012), and in a case study on the volcanic eruptions of Kasatochi and Sarychev CTM simulations are included, to analyse whether the measured MIPAS $SO_2$ after the eruptions can lead to the enhancements as seen in the MIPAS $H_2SO_4$ (qualitatively and quantitatively), and to study the transport patterns of the volcanic plumes.

The sentence will be replaced by the following: "Here, we present a new data set of sulfate aerosol volume densities (AVDs) retrieved from MIPAS measurements

(also converted into $H_2SO_4$ VMRs). The data are compared to MIPAS $SO_2$ and in a case study on two volcanic eruptions the MIPAS $H_2SO_4$ and $SO_2$ data are complemented by Chemical Transport Model (CTM) simulations. Analyses were made in terms of mass and transport patterns, to investigate the consistency of the MIPAS data sets and the evolution of volcanically emitted sulfur."

- Delete 'This paper has several purposes.'

  Thank you for this suggestion. Rather than deleting this sentence, we decided on better identifying the main purposes.

  The following changes will be made: "This paper has several purposes': (i)' we introduce a new data set of aerosol volume densities, retrieved from MIPAS measurements in Sect. 3, and '(ii)' compare the data to independent measurements of aerosols. We further study the distribution of MIPAS sulfate aerosol (as VMRs) in the period 2005 to 2012 and '(iii-b)' compare it to MIPAS $SO_2$. In Sect. 4 we perform (ii) a case study for two of the largest volcanic eruptions of the last decade in Northern Hemisphere mid-latitudes, which were measured by MIPAS. The volcanoes are Kasatochi (52.2° N/175° W) that erupted in August 2008, and Sarychev (48.1° N/153.2° E), which erupted in June 2009. In the case study we analyse MIPAS observations of $SO_2$ and stratospheric sulfate aerosol in comparison to CTM simulations, and study the sulfur mass contained in $SO_2$ and sulfate aerosol, together with the transport of their volcanic plumes. Finally, in Sect. 5 we draw last conclusions on the (iii) general consistency ..."

- Page 3, ln 3: Why did the authors choose the 2005-2012 period?

  The first period (Jun 2002–Mar 2004) is not considered within this study, as we aimed at investigating two of the major mid-latitudinal eruptions (Kasatochi in 2008, Sarychev in 2009) during the MIPAS measurement period (Jun 2002–Apr 2012). Volcanic eruptions during the first period only injected $SO_2$ masses of below 100 Tg to 10–22 km (Höpfner et al., 2015). Furthermore, the much

longer second measurement period (Jan 2005–Apr 2012) is characterised by a better vertical and horizontal resolution due to the denser vertical and horizontal sampling. For $SO_2$ data are available for the entire MIPAS measurement period. Future work will be invested into the retrieval of aerosol data from the first period in order to get an aerosol data set covering the whole MIPAS lifetime.

The revised version of the paper will include the following sentences on page 3, ln 16: "Here we concentrate on the data from the second and longer measurement period (Jan 2005–Apr 2012), as the major mid-latitudinal volcanic eruptions between 2002–2012 occurred during this period. Furthermore, this measurement period is characterised by an improved vertical resolution, especially in the altitude region of the upper tropopshere and lower stratosphere."

- Page 3: 'We analyse MIPAS observations of $SO_2$ and stratospheric sulphate aerosol in comparison to CTM simulations, and study the sulphur mass contained in $SO_2$ and sulphate aerosol, together with the transport of their volcanic plumes.'

This sentence needs to be re-worded. It is not clear that this analysis is related to the volcanic eruptions stated in the sentence before.

Thank you for pointing that out.

In the revised version it will be made clear that the sentence is related to the case study: "'In the case study we' analyse MIPAS observations ..."

- 'Finally, in Sect. 5 we draw final conclusions on the consistency between the MIPAS $SO_2$ and the new MIPAS sulphate aerosol dataset,'

Does this only relate to the data sets during the volcanic eruptions? Either replace 'Finally' or 'final'.

This sentence does not only relate to the presented case study, but to the entire data sets. It includes the comparison between the two MIPAS data sets, the comparison to the in situ data, and the model data.

The sentence is changed as follows: "Finally, in Sect. 5 we draw last conclusions on the general consistency between the MIPAS $SO_2$ and the new MIPAS sulfate aerosol data set, in combination with our model results in the case of the two volcanic eruptions, and give a short summary of our findings."

- Page 3, ln 23: It would be good if the authors could state the time period of the $SO_2$ data set they are using in this study so that the reader doesn't need to look into Höpfner et al. (2015).

  The information about the time period studied here can already be found on page 3, ln 16–17 "Here we only study data from the second measurement period. During this period, from January 2005 to April 2012, ..."

- Page 3, ln 28: Replace 'bias of' with 'bias between'

  This will be changed in the revised version.

- Page 3, ln 30: Replace 'within' with 'between'

  This will be changed in the revised version.

- '... when due to aerosol-related sampling artefacts the total mass of $SO_2$ was found to be strongly underestimated (Höpfner et al., 2015).'

  Re-word to: 'when the total mass of $SO_2$ was found to be strongly underestimated due to aerosol-related sampling artefacts (..).'

  This will be changed in the revised version.

- 'Their study comprises a dataset of volcanically emitted $SO_2$ for 30 volcanic eruptions, as seen in the MIPAS measurements'

  Not clear what 'their' refers to (I assume that is the study by Höpfner et al.). Suggest to reword this sentence and also to write 'as observed by MIPAS' rather than 'seen in the MIPAS measurements'.

To clarify that the referenced study is the study by Höpfner et al. (2015), the sentence will be changed to "The study by Höpfner et al. (2015) comprises a data set of volcanically emitted $SO_2$ for 30 volcanic eruptions, as observed by MIPAS.".

- Should Section 3 be moved forward to 2.1.3? The structure of this paper is not clear to me. Section 2 was labelled 'Data sets and Methods' but then Section 3 is 'MIPAS aerosol data set'. That is confusing.

  During the writing process of this paper Sect. 3 changed place. It was first included in Sect. 2, as you suggested. However, we then decided on separating the description of the new MIPAS data set from the general description of the already available data sets used in this study, and dedicateed a separate section to the new data. As one of the main purposes of the paper is to present the MIPAS aerosol data, and due to the length of the current Sect. 3, this structure seems more convenient to us.

  To make the separation into the description of available data sets in Sect. 2 and the new aerosol data set (Sect. 3) clearer, we rename Sect. 2 "2 Available observational data sets and model description" and Sect. 3 in "3 The new MIPAS aerosol data set"

- Page 4: 'The in situ measurements were completed with balloon-borne University of Wyoming optical aerosol counters and consist of size resolved aerosol concentrations from the surface to approximately 30 km.'

  Replace 'completed' with 'made' ... Suggest to change sentence to: "Size resolved aerosol concentration measurements from the surface to approx. 30 km altitude were made with ..."

  This will be changed in the revised version.

- Page 4: Reword sentence 'The latest style of the three was used initially in

2006, became the standard Laramie instrument in 2008, and was flown on quasi-Lagrangian balloons in Antarctica in 2010 (Ward et al., 2014).'

Something is missing in this sentence.

The sentence will be updated as follows: "The latest style (Laser based counters, LPCs) of the three instrument types was used initially in 2006, became the standard Laramie instrument in 2008, and was, as an example, also flown on quasi-Lagrangian balloons in Antarctica in 2010 (Ward et al., 2014)."

- Page 4, ln 12–13: 'For the MIPAS validation the Wyoming measurements were confined to those made with this final instrument (Laser based counter, LPC), which measures particles with radii $> 0.08$–$4.2$ $\mu$m in eight size classes."

Why were the data from this LPC used? Why not any measurements from other OPC measurements before 2006?

When profiles from the two types of instruments (LPC and WPC) are available for the same day, the retrieved in situ aerosol volume density profiles show differences below 20 km, with volumes estimated from the WPC generally larger by up to a factor of two. In general the MIPAS de-biased volume density profiles show good agreement with the in situ volume density profiles retrieved from both instruments above 20 km, but below 20 km the agreement is better with the LPC. A comparison of the differences between the LPC and WPC size distribution retrievals and a judgement as to their accuracy is beyond the focus of this study. We have chosen to use the in situ instrument which permits the simplest altitude dependent de-biasing function. Before 2006 no co-incident measurements could be found to the available WPC measurements, as MIPAS measurements were relatively sparse in 2005.

The manuscript will be changed as follows (page 4, ln 5–22): "To validate the new MIPAS aerosol data set described in Sect. 3, we use aerosol volume density profiles that were derived from in situ measurements of stratospheric aerosol

above Laramie, Wyoming (41° N, 105° W) (Deshler et al., 2003). Size resolved aerosol concentration measurements from the surface to approximately 30 km altitude were made with balloon-borne University of Wyoming optical aerosol counters. Measurements usually occurred between 6 and 9am, local time, with measurement frequency varying from monthly to bi-monthly. Data are available from 1971 to present. Over this time period three different primary instrument types were used. The latest style (Laser particle counters, LPCs) of the three instrument types was used initially in 2006, became the standard Laramie instrument in 2008, and was, as an example, also flown on quasi-Lagrangian balloons in Antarctica in 2010 (Ward et al., 2014). While the transition from the first instrument to the second was documented in Deshler et al. (2003), a similar study to compare the third Wyoming instrument with the second instrument is a work in progress. For the MIPAS validation, measurements from the second and third Wyoming instruments were available. The positive bias of MIPAS aerosol volumes from the in situ measurements was generally consistent between MIPAS and both of the Wyoming instruments above 20 km. Below 20 km the in situ measurements diverged from each other, with the second instrument indicating higher volumes than the LPC (third instrument), and at times higher than MIPAS. Based on these comparisons with both instruments the Wyoming measurements to be used were confined to those made with the LPC because it permitted the simplest altitude dependent de-biasing function for the MIPAS aerosol volume densities. The LPC measures particles with radii $> 0.08$–4.2 $\mu$m in eight size classes."

- Page 4, ln 15 and following: 'To derive geophysical quantities from the size resolved aerosol concentration measurements it requires fitting a size distribution to the data. In the past this has been done by choosing a subset of the measurements to fit either a unimodal or bimodal lognormal size distribution. The final size distribution selected is from that subset of the measurements which minimizes the root mean square error when the fitted distribution is compared to all the measurements. This approach has recently been changed to use laboratory measurements of the counting efficiency at each channel and then search the lognormal parameter space for the lognormal coefficients, which minimizes the error of the fitted distribution compared to the measurements.'

How does the 'new' approach compare to the 'old' approach? How different are the measurements when derived with the new method compared to the old approach? And why is the new approach not used but mentioned here?

As this paper is not intended to study the differences between retrieved profiles following the 'old' and 'new' approach we refer to a future paper on the newly retrieved data. In the profiles tested during the preparation of the manuscript (Laramie, Wyoming; Jan 2005–Apr 2012; with co-incident MIPAS locations) in single cases large differences arose between the 'new' and 'old' profiles, while overall the approaches show similar results, particularly for the LPC. The in situ data used in this manuscript were retrieved using the new approach.

On page 4, ln 21 the following paragraph will be substituted for the current text: "Deriving geophysical quantities from the size resolved aerosol concentration measurements requires fitting a size distribution to the in situ data. In the past this has been done by fitting either a unimodal or bimodal lognormal size distribution to a subset of the measurements. The final size distribution parameters selected are those from that subset of the measurements which minimises the root mean square error when the fitted distribution is compared to all the measurements. This approach is transitioning to a new approach which modifies the nominal in situ aerosol sizes based on laboratory measurements of the aerosol counting efficiency. The counting efficiency at each size is then included in a search of the lognormal parameter space for the lognormal coefficients which minimise the error of the fitted distribution, coupled with the counting efficiency, compared to the measurements. In our study we use the volume density profiles that are derived from the fitted lognormal size distributions (unimodal or bimodal, following the new retrieval approach) to the measurements. The precision of these volume estimates is the same as the old method, $\pm40$ % (Deshler et al., 2003). The change in the way the fitting parameters are derived is the subject of a paper to be submitted soon. The impact on size distributions from the LPC measurements is not large."

- Page 4, ln 24: 'The isentropic Chemical Transport Model used in our study ...'

Delete 'isentropic'. It is a CTM using isentropic levels as vertical coordinates but does that mean it could not be used differently? Include CTM acronym.

The version of the CTM used in this work does not comprise any other than isentropic levels as possible vertical coordinates, and is therefore suitable for stratospheric studies.

We will update the text as follows: "The Chemical Transport Model (CTM) used in our study (e.g. Sinnhuber et al.; Kiesewetter et al., 2010) is forced ... . 'The model uses isentropes as vertical coordinates.' "

- Page 4, ln 30: What are the initial OCS concentrations and where were they taken from?

As described in Sect. 2.3, the model simulations consider as only sulphur source the volcanically emitted $SO_2$ from Kasatochi and Sarychev, in individual simulations. Hence, OCS is not considered, as no background $SO_2$ formed from OCS is included in the simulations. However, OCS is part of the newly implemented sulfur scheme, and has therefore been mentioned here.

The following information will be included on page 5, ln 2: "... individually. 'Therefore, OCS is not considered as sulfur source in the simulations presented in this work.' "

[Figure]

It would help the reader if the chemical reactions included in the CTM would be listed somewhere. Does the model include liquid and gas-phase chemistry? Are you only considering chemistry in the stratosphere?

The only chemical reaction considered in this study is the reaction of $SO_2$ with OH, based on OH concentrations from a previous full chemistry run, and reaction rates from JPL (as described in Sect. 2.3). The model as used in this work does not include liquid and gas-phas chemistry and the implemented chemistry is applied throughout the entire model domain.

On page 4, ln 31 "In the sulfur scheme no distinction between tropospheric and stratospheric air is implemented." will be added. On page 4, ln 5 "The sulfur released from volcanic $SO_2$ reacts with OH (hydroxyl radical) to form $H_2SO_4$. 'This is the only chemical reaction considered in the simulations presented in this study.' "

You state 10 to 55 km, and depending on where you are on the globe, 10 km could still be in the upper troposphere. So does cloud uptake of $SO_2$ play a role here? If not, why not?

We did not consider any scavenging of $SO_2$ (neither $H_2SO_4$) in our model simulations. We expect the influence to be low in the altitude region studied here. In the tropics where the CTM reaches relatively low altitudes washout might have an impact on the sulfur distributtion. However, as can be seen in Fig. 8, in the troposphere MIPAS shows higher sulfur amounts than the CTM, even without us considering any losses due to precipitation. Furthermore, we intended to study the sulfur that enters the stratosphere, and as the MIPAS data show mostly the 'residual' $SO_2$' that remains in the atmosphere (above ~10 km) after the first weeks after the eruption, we expect the influence of cloud uptake to have a minor impact, if any.

On page 5, ln 17 we add information concerning the washout by precipitation: "In our simple sulfur scheme, no scavenging of $SO_2$ or $H_2SO_4$ by clouds is considered in the model. This would be confined mostly to tropospheric altitudes and in our study region ($\geq$ 10 km) especially to tropical latitudes. Washout by precipitation might play a role there but is expected to have a minor effect on our study, as we analyse the sulfur that remains in the atmosphere (above $\sim$10 km) after the first weeks following the volcanic eruptions."

- Page 5, ln 20: Do the authors believe that it will be clear to the reader what MIPAS/Balloon is in this context, especially since it is not mentioned before? I would suggest to add more information on this here and also to refer to it at MIPAS-B.

  To clarify the difference between MIPAS/Balloon and MIPAS/Envisat, the text will be updated as follows: "In previous analyses of mid-infrared observations by MIPAS-B (the balloon-borne predecessor of the MIPAS satellite instrument; Friedl-Vallon et al., 2004) and MIPAS/Envisat (MIPAS instrument on the satellite Envisat, generally referred to as "MIPAS" throughout the present work) it has been demonstrated ..."

- Page 6, ln 2: Remove 'E.g.'

  This will be deleted in the revised version.

- Figure 3: Is it possible to show the uncertainties on the in situ measurements? What does LCP 2m, 1p and 3m stand for? The standard deviations on the mean values are shown but that is not the uncertainty on the measurement which would be interesting to have a look at? Wouldn't it be more appropriate to calculate the uncertainty on the mean using the measurement uncertainties on each datum that went into the mean calculation?

  As noted in the manuscript (page 4, ln 21), the precision of the aerosol volume densities is given as $\pm$40 %. Such error bars will be added to Figure 3 for the LPC measurements. As we only have this rough and non-variable estimate of

the precision of the in situ data, it has not been considered in the calculation of the mean. LPC 2m, 1p and 3m are different instruments of the same type. These distinctions are unimportant and will be removed from the legend. All the in situ measurements will be designated as LPC.

- Page 10, ln 3: 'In Fig. 4a the standard errors of the mean show the uncertainty of the bias.'

  I don't understand this statement. How does a standard error of the mean profile relate to the uncertainty of the bias? This needs to be clarified.

  In Fig. 4a the standard errors of the mean profiles are shown, both for the in situ data and MIPAS data. The magnitude of these uncertainties provides information on the statistical uncertainty of the bias (difference between the mean in situ and MIPAS profile, magenta and blue profiles in Fig. 4a). To clarify what we intended to say we will include the statistical uncertainty of the bias, calculated based on the standard errors shown in Fig. 4a in Fig. 4b. This uncertainty of the difference between the in situ and MIPAS data is calculated as follows: $\sqrt{x_1^2 + x_2^2}$, with $x_1$ and $x_2$ being the standard errors at a specific altitude.

  The manuscript will be changed as follows: Page 10, ln 3: "In Fig. 4a the standard errors of the mean profiles are presented, and in Fig. 4b the statistical uncertainty of the bias (difference between the mean in situ and mean MIPAS profile) is shown." Page 10, ln 14: "In (b) the statistical uncertainty of the absolute difference between in situ and MIPAS data is shown (horizontal pink lines; square root of the sum of the 1-sigma standard errors squared for MIPAS and the in situ data)." Page 11, ln 12: "... also suits well. 'The uncertainty of the bias (Fig. 4b) at altitudes above ∼17 km shows that the positive bias is not random, as the spread is low and uncertainty limits are noticeably distant from zero."

- Figure 3 shows a strong signal at around 18 km in the balloon measurements on 28.07.2011 which does not seem to reflect in any way in the mean and its

standard error shown in Figure 4a. Is that expected? This takes me back to the point that the mean values and their standard error do not take into account the uncertainty on each measurement.

Thank you very much for making us realise that unfortunately information is missing on the data Fig. 4 is based on. In Fig. 3, the profiles measured on the 28.07.2011 are presented, but in Fig. 4 the profiles were calculated neglecting this day. During the work done in preparing the manuscript both calculations, including and excluding that day have been made to test how they change the mean profiles and the de-biasing, as on this day profiles show large variability between MIPAS and the in situ data at low altitudes, and strongest vertical variability of all in situ profiles ($<$ 18 km). As the de-biasing is based on altitudes from 18–30 km, it basically does not change when ex- or including that day. This is shown by this figure:

The text will be updated as follows: Page 10, ln 1: "... mean over the profiles that were retrieved from LPC measurements, as shown in Fig. 3 (excluding the 28.07.2011) ..." Page 10, ln 3: "The profile on the 28.07.2011 shows large differences between MIPAS and the in situ data and the strongest vertical variability of all in situ profiles at low altitudes (below ∼18 km), possibly due to the Nabro eruption (12 Jun 2011). Hence it is exluded from the calculation of the mean profiles shown in Fig. 4." Page 11, ln 16: "By excluding the in situ and MIPAS profiles measured on 28.07.2011 in the calculation of the mean profiles, the agreement between the measurements is improved in the altitude range below 18 km, while above this altitude changes are marginal, as can be expected from Fig. 3. The de-biasing is therefore not affected by the dismissal of the observations from this day." In the caption of Fig. 4 the following will be added: "... data in Fig. 3 '(excluding the 28.07.2011)', ..."

- What did the authors do to de-bias MIPAS measurements shown in Figure 3 and 4? There are some words around it on page 10 (ln 5 to 10) but this didn't answer

the question about what the authors actually did.

The difference of the mean MIPAS profile to the mean in situ profile (both shown in Fig. 4a, as pink and blue solid lines) has been calculated, resulting in the pink solid line in Fig. 4b. The linear least squares fit to this latter profile (fit to 18–30 km, pink dashed line) represents the vertically resolved values of the de-biasing. For the de-biasing each MIPAS profile was reduced by the corresponding aerosol volume densities ($\sim$0.075 $\mu$m$^3$cm$^{-3}$ at 10 km, $\sim$0.025 $\mu$m$^3$cm$^{-3}$ at $\sim$27 km).

Page 11, ln 10–11 will be changed to "The de-biasing is based on the absolute differences between the aerosol volume densities of the mean MIPAS and in situ profiles (Fig. 4b, pink solid profile) at 18–30 km, where profiles show weak variability and low uncertainty of the bias. A linear least squares fit (Fig. 4b, pink dashed line) to the profile of absolute differences represents the vertically resolved values of the de-biasing, which are subtracted from each MIPAS profile during offset-correction."

- Page 13, ln 1: 'Sulphur is released from OCS mainly in the tropics at altitudes between about 25 and 35 km (Brühl et al., 2012) and the sulphate aerosol that is built is transported towards mid-latitudes and lower altitudes.'

  Is that statement still true with new publications (e.g. Lennartz et al. (2017)) about OCS being published?

  We were not refering to emissions of sulfur from OCS at tropospheric altitudes close to the surface but to stratospheric altitudes.

  In the revised version this information will be added (" 'In the stratosphere' sulfur is released ...")

- Page 13: 'In the mid-latitudes of the Northern Hemisphere, the sulphate aerosol is increased during boreal summer at around 10–12 km.'

I'm not sure if that is a strong enough signal as the Northern Hemisphere is strongly disturbed by volcanic eruptions according to Figure 5. What could be a cause for this increase?

To a certain extent this signal can be distinguished in each year in which volcanic influence at these altitudes is low, most clearly visible in 2005–2007, both in the MIPAS $SO_2$ and aerosol data. Our CTM is not suitable for studying possible causes of these enhancements. We do not expect the signal to be caused by volcanoes, as we see it on annual basis. It might be due to strong upwelling of polluted air (anthropogenic / wild fires). However, these are speculations and are therefore not included in the manuscript.

- Figure 5: There is a strong signal at altitudes 18 to 22 km (around 2007), i.e. enhanced $H_2SO_4$ concentrations which cannot be seen in the $SO_2$ concentrations. Do the authors have an explanation for this signal in the tropics that high up?

This signal in the aerosol data is caused by upward transport of sulfate aerosol that has been formed from volcanic $SO_2$ (tropical volcanoes such as Soufrière Hills and Rabaul). Vernier et al. (2011) studied SAGEII and CALIPSO data for $20°$ S–$20°$ N showing the upward transport of aerosol, and a similar pattern is present in the MIPAS $H_2SO_4$ volume mixing ratios. The MIPAS data of $SO_2$ are enhanced in the tropics at comparable altitudes as the aerosol data, but only in the beginning after the eruptions, as the removal of $SO_2$ by chemical reaction with primarily OH is way faster than the removal of sulfate aerosol. Due to its faster removal it does not show a similar upward motion, as seen in the aerosol data.

On page 11, ln 26 this information will be added: "... above 16 km. The aerosol is lifted upwards with time and the plumes get modulated by the Quasi-Biennial Oscillation in the tropics. A similar pattern of upward motion of the volcanic aerosol from these tropical eruptions has been seen in satellite measurements of aerosol extinction ratios (Vernier et al., 2011)."

- Page 14, ln 8: Is 'sedimentation radius' the right term to be used? I don't think it is (I think the authors mean the radius of the aerosol) and the authors might want to think about rewording this sentence and where appropriate throughout the paper.

  In the CTM we use a constant aerosol radius to determine the terminal fall velocity, intending to simulate sedimentation with a constant average settling velocity that corresponds to aerosol of different radii.

  We will name the simulation radius "effective sedimentation / settling radius" throughout the paper and add the following information on page 5, ln 17: "In the atmosphere the radius of sulfate aerosol varies (Deshler et al., 2003 and 2008). Nevertheless, for simplification we use a constant "effective sedimentation radius" to determine the terminal fall velocity, which we consider to be the average settling speed of aerosol particles of different radii."

- Page 14, ln 9: 'Good accordance between the modelled and measured $SO_2$ masses is essential to test, ...'

  This sentence is rather odd and 'accordance' should be replaced with 'agreement'?

  The sentence will be changed as follows: "As we intend to test if the measured aerosol is quantitatively and qualitatively consistent with its measured precursor by comparison with modelled sulfate aerosol, a good agreement between the modelled and measured $SO_2$ masses is essential."

- Page 14, ln 11: 'In Table 1, $SO_2$ masses for three altitude regions ...' Include 'injected $SO_2$ amounts for three altitude regions as used in the CTM simulations ...' The authors do not explain why and how they have chosen these amounts. Please clarify.

  Your suggestion will be included in the revised version. The injected $SO_2$ amounts have been chosen from a sensitivity study with differing injected masses (partly

based on masses as given by Höpfner et al., 2015), with the intention to achieve good agreement between the modelled and measured $SO_2$ (agreement starting approx. one month after the respective eruption; comparison of masses as in Fig. 6). The presented $SO_2$ masses resulted in the best agreement.

This will be clarified on page 14, ln 14: "Simulations have been made with varying injected $SO_2$ masses and upper injection altitude limits, intending to achieve good agreement between the modelled and measured $SO_2$ masses (comparisons as in Fig. 6). The data presented here resulted in the best agreement, with comparisons starting approximately one month after the respective eruption (explanation in the following)."

- Page 14, ln 12: 'The simulations result in good agreement between measured and modelled $SO_2$.'

Can the authors please clarify what simulations they are talking about? What is the simulation period? What the time step? I feel that there are more information required about the CTM simulations.

We are referring to the simulations as presented in Fig. 6 (orange and blue lines). The simulations were made with a time step of 30 min, started on the last day of the month preceeding the eruption, espectively, and covered 365 d. As the background for $SO_2$ and $H_2SO_4$ has been set to be zero throughout the entire model domain, and the model is driven by ERA-Interim data that are read in every 6 h, no real spin up time is needed. As the mentioned sentence is related to a Figure that is only described later (Fig. 6), it will be deleted here.

Instead, on page 17, ln 18 we add the following sentence: "After this first month, the simulated $SO_2$ agrees well with the measurements by construction." In Sect. 2.3 "Chemical Transport Model" the following information will be added (page 5, ln 17): "The model is run for 365 d per simulation, with a time step of 30 min and tracer fields are written out daily at 12 UTC. For the eruption of

Kasatochi (7 Aug 2008) the individual runs are started on the 31 Jul 2008, and for the eruption of Sarychev (12 Jun 2009) all runs are started on the 31 May 2009. As the initial tracer fields are set to zero and the model is driven by ERA-Interim reanalysis data, which are updated every six hours, no long spin up time is needed. Per volcano four simulations were made that differ concerning the particle size of sulfate aerosol. Simulations were made with constant aerosol radii of 0.1, 0.5 and 1 $\mu$m, and without sedimentation."

- Page 14, ln 14: 'This method was applied as in the first month after the eruption MIPAS underestimates the $SO_2$ (Höpfner et al., 2015).'

I don't understand what the authors are trying to say here. This method was applied in their study or the study of Höpfner et al., (2015). Why is it important to know how Höpfner et al. derived the mass? How did the authors derive the $SO_2$ mass used for their study? Could the authors please clarify why they didn't use the same values as Höpfner et al. and how they derived their values for $SO_2$ mass injected by the two volcanoes? And why do they not provide any uncertainties on their values?

For parts of your questions we refer to a previous question where we answered how the injected $SO_2$ masses were derived in our study. The method mentioned here was applied in the study by Höpfner et al. (2015), not in our study. However, as the injected $SO_2$ masses are partly based on the study by Höpfner et al. (2015) we think that is is valuable to include this information in our manuscript. For both volcanoes the simulations resulted in better agreement with the measurements when injecting less $SO_2$, compared to the masses given by Höpfner et al. (2015). For Kasatochi the masses by Höpfner et al. (2015) minus their given uncertainties are used, which are relatively large, due to the method they applied to derive the masses. The number of simulations we made was not sufficiently large to calculate uncertainties for the presented $SO_2$ masses.

The manuscript will be changed as follows: "They applied this method as in the

first month after the eruption MIPAS underestimates the $SO_2$ (Höpfner et al., 2015). Their method results in relatively large error bars that depend on the time period the fit is based on (Höpfner et al., 2015; presented also in Table 1)." On page 14, ln 14 the following information will be added: "Due to the limited number of simulations no uncertainties are given for the presented $SO_2$ masses. The main part of $SO_2$ ..."

- Figure 6: Wouldn't a sedimentation radius of 0 mean that there are no aerosols? Do the authors mean that aerosol doesn't get lost through sedimentation?

  This was only an internal label indicating "no sedimentation" - we will correct this.

  The statement on page 16, ln 4–5 will be corrected to: "For the CTM four simulations for $H_2SO_4$(liq) are shown with different effective sedimentation radii (0.1, 0.5, and 1 $\mu$m), and without sedimentation." On page 15, ln 8–9 the statement will be changed to: "The simulations differ concerning the implemented sedimentation (no sedimentation and effective sedimentation radii of 0.1, 0.5, and 1 $\mu$m)."

  I don't understand why the eruption of Redoubt (23 Mar 2009, 60.5° N/152.7° W) is included here but was never mentioned before? Also, it is not clear if additional $SO_2$ has been injected in the model due to this eruption (I suspect not). If the authors want to include Redoubt in the figure because there is a signal in the measurements, they should state that clearly in the text.

  No additional $SO_2$ has been injected in the model due to the eruption of Redoubt. It is mentioned here, as it can be seen in the MIPAS measurements.

  To clarify why it is indicated in Fig. 6, on page 15, ln 11 the following information will be added: "In Fig. 6, the eruption times of Kasatochi (7 Aug 2008) and Sarychev (12 Jun 2009) are indicated. Additionally, the eruption time of Redoubt (23 Mar 2009) is marked, as this eruption produces a signal in the measurements. It is not included in the simulations, however."

What in the CTM causes $SO_2$ to be lost so much faster between 10.5 and 14.5 km compared to 14.5 and 18.5 km? Again, I think more information about the CTM is required.

The possible reasons for loss of $SO_2$ in the model from a confined altitude and latitude region are due to chemical loss or to vertical or horizontal transport out of the region. As noted in Sect. 2.3, the vertical transport is calculated based on ERA-Interim heating rates, the horizontal transport based on ERA-Interim wind fields, and for the chemical lifetime the concentration of OH and the ambient temperature and pressure are important. We did not study which process dominates the removal in the different altitude regions shown in Fig. 6.

On page 15, ln 11 the follwing information will be added: "In the model the two species can be removed from one confined altitude and latitude region due to transport (advection: $SO_2$, $H_2SO_4$; sedimentation: $H_2SO_4$) or chemical loss ($SO_2$)."

The possible reasons for loss of $SO_2$ in the model from a confined altitude and latitude region are due to chemical loss or to vertical or horizontal transport out of the region. Loss of sulphate aerosol in the model is possible due to sedimentation and advective transport. The sum of $SO_2$ and sulphate aerosol neglects the chemical loss and comparisons between simulations with and without sedimentation show the influence of sedimentation on the removal. A comparison of Fig. 6 (a) and (d) reveals a fast decay of the total sulfur in the case of Kasatochi, which is mostly connected to advection (removal in the Brewer-Dobson circulation), as the differences between the simulations with and without sedimentation are low and the fast decay is present both in the curve of $SO_2$ and total sulfur. Only after some months the effect of sedimentation increases, when sulfur from above reaches this altitude region (second peak). In the case of Sarychev the curve of $SO_2$ and total sulfur are rather different, as especially the sulfate aerosol is supplied by aerosol that is transported downwards from above. This can also

be seen in Fig. 7.

Can the authors explain why for a radius of 1 $\mu$m the $H_2SO_4$ loss is levelling off earlier (around November) between 18.5 and 22.5 km than compared to the other altitude ranges?

In the uppermost altitude range only little aerosol is present, and aerosol with a radius of 1 $\mu$m settles relatively fast, compared to smaller particles. Furthermore, at higher altitudes sedimentation velocities are higher. Therefore the relatively little excess amount of aerosol is removed rapidly.

On page 18, ln 8 the follwing information will be added: "... can be expected. Model simulations show that compared to 10.5–18.5 km and compared to small particles, the bigger particles level out faster in the uppermost altitude range studied here. Reasons for this faster removal of the volcanic aerosol are that only little aerosol is injected in the altitude region 18.5–22.5 km, that bigger particles settle faster, and that settling velocities rise with increasing altitude due to the corresponding decrease in air density."

What is causing the second peak (around Dec) in $H_2SO_4$ between 10.5 and 14.5 km? This peak is also seen in the MIPAS measurements but shifted by half a month. The peak in $SO_2$ as modelled by the CTM is shifted compared to the peak in MIPAS $SO_2$. Can the authors explain why that is?

This second peak is caused by downward transport of sulfur that has been emitted by the eruption of Kasatochi. It is described on page 19, ln 1–8. The time shift has not been studied in detail, but is assumed to be caused by model uncertainties and sampling artefacts, as in this altitude region both the measurements and model data are sparse.

We will mention the second peak on page 18, ln 25: "A peak can be seen in the measured and modelled sulfur dioxide and sulfuric acid masses in November / December 2008 (Fig. 6) in the lowermost altitude region (10.5–14.5 km). This

peak is caused by downward transport of sulfur in the extra-tropics that has been emitted by the eruption of Kasatochi. In the following section (Sect. 4.2) more details are given on this transport pattern."

- Page 18, ln 15: '... due to its reaction with OH, and sulphate aerosol is consequently built.'

This sentence needs to be reworded – use formed rather than build.

This will be changed in the revised version.

- Page 18, ln 23–24: '... the modelled sulphur mass without sedimentation already compares rather well with the measured sulphur mass.'

Looking at Figure 6d to f, are the authors really saying that the no-sedimentation run compares well to the measurements, especially in 18.5 to 22.5 km region? I do not agree with this statement and their conclusions drawn from this. The fit to the measurements is better for the runs where sedimentation was considered.

It is true that we need to clarify better why we say that the no-sedimentation run already compares well to the measurements. We are refering to the long-term decay of the sulfur mass. In the beginning sedimentation has a strong influence on the absolute amount of sulfur and the impact on the sulfur that still remains in the altitude ranges after a long period of time is clearly visible, but the long-term decay is quite similar in all simulations. The long-term removal is not dominated by sedimentation but by transport in the Brewer-Dobson circulation. Furthermore, the upper altitude range where sedimentation is fastest has no strong impact on the absolute sulfur mass and its removal as only a minor amount of sulfur is found at these altitudes.

On page 1, ln 20 will be changed to: "While sedimentation of sulfate aerosol plays a role, we find that the long-term decay of stratospheric sulfur after these volcanic eruptions at mid-latitudes is controlled mainly by transport in the Brewer-Dobson

circulation." On page 18, ln 19 the following will be added: "This becomes obvious when comparing the 'long-term removal of' total modelled sulfur ..." On page 18, ln 23–24 this information will be added: "... the modelled 'decay of' sulfur mass without sedimentation already compares rather well with the measured 'decay of' sulfur mass." On page 24, ln 6 will be changed to "While sedimentation of sulfate aerosol does play a role, the 'decay' of sulfur in the mid-latitude lower stratosphere ..."

- Figure 7: Again why is the Redoubt eruption included in this Figure.

The eruption of Redoubt is indicated, as the MIPAS data show a signal caused by this eruption.

The caption of Fig. 7 will be updated as follows (page 19, ln 16): "Indicated are the days of the eruptions of the volcanoes Kasatochi in Aug 2008, Redoubt in Mar 2009, and Sarychev in Jun 2009, which were observed by MIPAS. Redoubt is not considered in the model." In the captions of Fig. 8 and 9 similar information is added.

This is the first time where the authors mention the simulation periods. This has to come earlier and it has to be described in the text.

On page 5, ln 17 the following information will be added: "The model is run for 365 d per simulation, with a time step of 30 min and tracer fields are written out daily at 12 UTC. For the eruption of Kasatochi (7 Aug 2008) the individual runs are started on the 31 Jul 2008, respectively, and for the eruption of Sarychev (12 Jun 2009) all runs are started on the 31 May 2009. As the initial tracer fields are set to zero and the model is driven by ERA-Interim reanalysis data, which are updated every six hours, no long spin up time is needed."

Why is the pattern for $H_2SO_4$ (CTM simulation) for the Kasatochi eruption so different from the Sarychev eruption? For the Kasatochi eruption the $H_2SO_4$ concentrations are lower than for the Sarychev eruptions although similar amounts

of SO$_2$ were injected?

One big difference between the eruptions of Kasatochi and Sarychev can already be seen in Fig. 6 and Table 1. In the case of Kasatochi a relatively large part of the SO$_2$ is injected at altitudes below 14 km (518 Gg or 77% of the injected SO$_2$) and transported downwards and out of the altitude region we study in the present work relatively fast. Therefore it is not reflected in the aerosol loading. In the case of Sarychev more of SO$_2$ is injected into the altitude region above 14 km (367 Gg or 48% of the injected SO$_2$). Hence, more SO$_2$ is available after the eruption of Sarychev to be converted into sulfate aerosol and to stay in the altitude range studied here for a relatively long period of time.

On page 19, ln 11 this information will be added: "Parts of the differences between the transport patterns after the eruptions arise from the injected SO$_2$ masses. In the case of Kasatochi the main part of SO$_2$ was injected to altitudes below 14 km (518 Gg or 77 % of the injected SO$_2$). It is transported downwards and out of the region studied here relatively fast and therefore only a minor part is reflected in the aerosol loading. In the case of the Sarychev eruption almost half of the SO$_2$ (367 Gg or 48 % of the injected SO$_2$) is injected into the altitude region above 14 km. It is available for conversion into sulfate aerosol for a longer period of time, as can be seen in the higher H$_2$SO$_4$ volume mixing ratios after the eruption of Sarychev, compared to Kasatochi."

How much of the simulated Kasatochi 'double-plume' is made due to the choice of the SO$_2$ mass being injected at different altitude levels?

Neither in a CTM simulation with the SO$_2$ mass from Kasatochi, which erupted on the 7 Aug 2008 at 52.2° N/175.5° W, injected on the day and at the location of the Sarychev eruption (12 Jun 2009, 48.1° N/153.2° E), nor in a CTM simulation with the Kasatochi SO$_2$ mass injected at the location of Sarychev (7 Aug 2008, 48.1° N/153.2° E), a comparable separation into an upper and lower part of the plume is simulated. When the SO$_2$ mass from Sarychev is injected on the day
and at the location of the Kasatochi eruption (7 Aug 2008, 52.2° N/175.5° W) no strong separation is simulated either. These CTM simulations therefore suggest that the pattern of the 'double-plume' is caused by the meteorological situation after the eruption in combination with the vertical distribution of injected $SO_2$.

On page 19, ln 11 this information will be added: "Model simulations with 'switched' $SO_2$ masses (mass of Kasatochi injected on the day and at the location of Sarychev, and vice versa), and a simulation with the $SO_2$ mass from the Kasatochi eruption injected at the location of Sarychev, reveal that the 'double-plume' that has been observed after the eruption of Kasatochi results from the combination of the vertical distribution of injected $SO_2$ masses and the prevailing transport after the 7 Aug 2008, the eruption date of Kasatochi, in the model driven by wind fields and heating rates. Neither of the simulations results in a comparable separation into an upper and lower part of the plume."

The injection heights of $SO_2$ for the Sarychev eruption seem not appropriate when compared with the measurements. The model simulations show high $SO_2$ concentrations from 10 to 19 km while in the observations the range goes from 11 to 16 km. Did the authors conduct any sensitivity studies regarding the injection height and the impact on their results?

When keeping in mind that MIPAS has problems in detecting $SO_2$ up to approx. one month after the eruption, we do not see such a strong mismatch between the altitude range covered by the measured and modelled sulfur dioxide after the eruption of Sarychev (right plume in each plot in Fig. 7). Concerning the injection height and the impact on our results, no extensive sensitivity assessment has been done. However, we made simulations with varying upper injection limits and concluded that 19 km was resulting in reasonable agreement between the measurements and simulations.

- Page 19: "and reaches 10 km after a few months."

The upper plume doesn't reach 10 km after a few months as shown in Figure 7. It is moving down but by doing so that sulfur concentration reduces. Or are the authors talking about the model simulations?

The concentrations are reduced during the downward transport of the upper part of the plume, and not the bulk but some parts of the plume reach 10 km after several months. It is true that only very few sulfur reaches 10 km.

This part of the sentence will be replaced by: "... circulation. During the descent the sulfur concentrations are reduced and some parts of the sulfur reach 10– 12 km after several months."

• Figure 8: The CTM model shows about 1 month lag before $SO_2/H_2SO_4$ is seen in the tropics compared to the observations which show a signal right from the beginning although the signal in the observations might not be from the volcanoes investigated here? The signal in the measurements in August must be from a different volcano?

As can be seen in Fig. 5 and has been noted in the manuscript, the MIPAS $SO_2$ and $H_2SO_4$ data show enhancements in the tropics at altitudes of about 14– 16 km during the entire measurement period (with varying intensities), which are supposed to only partly be connected to volcanic eruptions. A certain influence of elevated aerosol values due to cirrus clouds that have not been captured by the ice-filter is possible. We cannot quantify from the observations to which extent the enhancements seen in the measurements in Fig. 8 are caused by the eruptions of Kasatochi and Sarychev. It is clear that parts of the sulfur that has been observed from November 2008 to ∼February 2009 has been injected into the atmosphere by the eruption of Dalafilla (Nov 2008).

On page 20, ln 9, the following information will be added: "The relatively high values in the measurements at around 13–16 km have already been noted in Fig. 5 and are supposed to only partly be connected to volcanic eruptions. A

certain influence of elevated retrieved aerosol values due to cirrus clouds that have not been captured by the ice-filter (Sect. 3.1) is possible. To which extent the observed enhancements in the measurements (Fig. 8a–c) are caused by the eruptions of Kasatochi and Sarychev is not clear. In the case of Kasatochi model simulations suggest that enhancements are confined primarily to altitudes above approximately 16 km. Additionally to the tropical enhancements at 13–16 km, the eruption of Dalafilla in November 2008 overlays with the observed sulfur that has been emitted by Kasatochi. The CTM simulations of Sarychev indicate that sulfur observed at altitudes as low as 12 km can be attributed to the volcanic eruption."

For the Sarychev eruption, the model seems to show the signal in the tropics earlier than it is seen in the measurements. Why could that be? The lag (time lag for when the eruption is seen in the tropics) seems to be reduced for the Sarychev eruption? The model seems to overestimate the peak amount of $SO_2+H_2SO_4$ compared to the observations. Any explanation for that given that the model agrees with the peak values of measurements better between 30 and 90N?

Comparisons of daily global horizontal distributions of the measurements and model results show that the differences between the measurements and simulations seen in Fig. 8 are partly due to sampling artefacts and partly due to a different horizontal extent of the volcanic plumes. In some regions where the model data show enhancements, measurements were filtered out due to clouds (ice, water, ash), especially in the tropics ($SO_2$: up to around 14/15 km and in the region of the Asian monsoon up to around 16/17 km; aerosol: primarily up to around 18/19 km and in the region of the Asian monsoon up to about 20 km). This might produce a low bias in the zonal mean of the measurements, compared to the CTM data that covers the entire globe (above 13 km). Additionally, the horizontal extension of the plumes is not in perfect agreement between the measurements and observations, provoking further differences between the zonal mean volume mixing ratios. In the case of the Sarychev eruption a relatively strong meridional transport of the sulfur is simulated already in the first days after the eruption. Simulated $SO_2$ reaches 15° N 8 d after the eruption at 14 km and 16 km, and 12 d after the eruption at 18 km (with VMR $>$ 1,500 pptv). This strong meridional tranpsport is not reflected in the measurements – where available – leading to the time lag and higher sulfur content in the tropics in the modelled data.

On page 20, ln 9, the following information will be added: "Differences between the presented zonally averaged measurements and model results arise partly from the fact that MIPAS measurements are not uniformly distributed and data were filtered, and due to sparse data coverage in the case of the CTM up to an altitude of 12–13 km. Data are partly missing in relatively large areas, which may lead to biased zonal means. In the measurements, for $SO_2$ data are missing particularly in the tropics at altitudes below about 15/16 km and at higher altitudes (up to ∼17 km) in the region of the Asian Summer Monsoon. In the case of measured sulfate aerosol data are filtered especially in the tropics at altitudes up to about 18/19 km and in the region of the Asian Summer Monsoon (up to ∼20 km) and in polar regions entire profiles were filtered out due to PSCs. In the case of the CTM data coverage is reduced up to an altitude of 13 km. Especially after the eruption of Sarychev a higher sulfur content is simulated in the tropics compared to the measurements (Fig. 8) and enhancements are seen few days after the eruption. This results from a strong modelled meridional transport of $SO_2$ after this eruption. At about 12–16 km altitude the injected $SO_2$ reaches 15° N 7–8 d after the eruption. This strong southward transport early after the eruption is not reflected in the measurements, which are, however, partly missing in the tropics due to filtering."

- Figure 9: For the CTM simulations, it looks like $SO_2$ is immediately enhanced at lower latitudes (down to 15deg N) at the time of the eruption, especially for the Sarychev eruption. Why is that? $SO_2$ is much more confined to the NH in the

observations.

Due to the prescribed wind fields during the first days after the eruption, in the model the Sarychev plume is spread relatively fast in meridional directions. At that time only few MIPAS measurements show high values, and not that far South. Also after a longer period, the measurements (where available and not filtered out) do generally show less enhancements in the tropics than the CTM. The differences might be caused by model uncertainties or inaccuracies in the prescribed wind fields. Difficulties with the measurements / retrieved profiles are another possible reason for the differences. After the eruption of Kasatochi less of a latitudinal spread is simulated and the plume is rather compact in the beginning. It takes about 16 days for the plume to reach $15°$ N. At that time the MIPAS $SO_2$ already shows a relatively strong signal produced by the volcano, that extends to these tropical latitudes.

How is the $SO_2$ emitted by the volcanoes injected into the model? The authors state that '$SO_2$ is uniformly distributed to the grid boxes per altitude range' but then I don't understand why the picture for Sarychev at 12 and 14 km is so different from the picture for Kasatochi given that $SO_2$ was uniformly distributed between 10 and 14 km (just looking at the time of the eruption, realizing that things will change after some time)?

In the model the $SO_2$ is injected on the day of the eruption at 12 UTC, into the column of grid boxes that includes the location of the volcano. The mass for a given altitude range has been distributed equally to the air mass contained in the grid boxes for which the centres lie in this altitude range, excluding the lowermost (boundary) level. The differences in the meteorological situation / wind fields and in the injected sulfur dioxide masses result in the differences between the simulations of Kasatochi and Sarychev. The vertical distribution of grid boxes is rather similar at the eruption times and locations of Kasatochi and Sarychev, therefore, this can be excluded as explanation for the differences.

I find the differences in the latitudinal distribution of SO$_2$ between the observations and the model interesting and would like the authors to comment on that.

As noted before and included in the manuscript, sampling artefacts and differences in the horizontal extent / location of the plumes cause disagreement between the zonal mean values calculated from simulations and observations.

- Page 22, ln 20: 'Especially at low altitudes we find a mixing barrier at$\sim$30° N, with a strong gradient between low values in the tropics and high values in the extra-tropics, which weakens towards higher altitudes.'

Which seems to be more pronounced in the measurements than in the CTM, especially for SO$_2$. Do the authors have any explanation for the 'leak' towards the equator in the model, especially in July/Aug after the Sarychev eruption?

As noted before sampling artefacts and differences in the horizontal extent / location of the plumes cause disagreement between the zonal mean values calculated from simulations and observations. In the case of the Sarychev eruption a strong meridional transport is simulated early after the eruption, which cannot be proved by comparisons to the MIPAS measurements. Partly due to missing data in the region of high modelled sulfur content and partly due to elevated simulated sulfur amounts in regions of low measured VMRs.

- Page 22, ln 25: 'An additional transport process starts at an altitude of about 18 km in the case of Kasatochi and $\sim$16 km in the case of Sarychev (Fig. 9 and 10).'

What 'additional' transport process are the authors referring to here?

This sentence refers to the southward transport of sulfur.

For clarification the manuscript will be changed to: "At an altitude of about 18 km in the case of Kasatochi and $\sim$16 km in the case of Sarychev, and at altitudes above, a southward transport of sulfur is noticed (Fig. 9 and 10)."

[Figure]

- Figure 10: Enhanced $H_2SO_4$ concentrations reach further South in the model simulations than the observations show (at 18 km). In the observations $H_2SO_4$ is more confined to latitudes between 15 and 60deg N while for the model simulations, enhanced $H_2SO_4$ concentrations reach the equator. Why the difference?

  In the tropics at an altitude of 18 km the data coverage for MIPAS $H_2SO_4$ is relatively sparse. In the case of Sarychev it is not clear if a better data coverage would improve the agreement or if it is a 'real' difference between the measurements and the model results, which show enhancements that reach the equator. After the eruption of Kasatochi the modelled $H_2SO_4$ is transported southwards but elevated values do not reach that far towards the equator. Therefore a possible influence of filtered MIPAS data on the differences is weaker.

- Page 23, ln 12: 'The weaker southward transport in the case of the Kasatochi eruption that starts at higher altitudes, compared to the Sarychev eruption, could be due to the eruption having been later during the monsoon season, leading to enhanced southward transport by the Asian summer monsoon anticyclone for a shorter period of time.'

  This sentence is confusing ... 'weaker transport leading to enhanced transport'? Could the authors please clarify this sentence. I think I know what the authors mean but they need to clarify which eruption they are talking about in the second part of the sentence 'could be due to the eruption'.

  This sentence will be changed to: "Compared to Sarychev, the southward transport of the Kasatochi eruption plume is weaker and initiates at higher altitudes. This might be explained by the eruption of Kasatochi having been later during the monsoon season, resulting in a shorter time period of enhanced southward transport induced by the Asian summer monsoon."

- Page 23, ln 20: 'In this study a new dataset of MIPAS/Envisat global aerosol volume densities and liquid-phase $H_2SO_4$ VMR distributions is presented for 2005

to 2012 ...'

Are the authors not talking about two data sets that are presented? Replace 'is' with are and use data sets instead of dataset.

We are actually talking about one data set of aerosol volume densities that has been converted into $H_2SO_4$ volume mixing ratios.

The sentence will be changed as follows: "In this study a new data set of MI-PAS/Envisat global aerosol volume densities, also converted into liquid-phase $H_2SO_4$ volume mixing ratios, is presented for ..."

- Page 24: 'The new $H_2SO_4$ aerosol observations enable us to further constrain the total sulphur emitted into the stratosphere by the Kasatochi and Sarychev eruptions and to revise our previous estimates that were based on $SO_2$ observations only.'

So what is the new estimate of total sulfur emitted into the stratosphere by both volcanoes? It would make sense to include that in the conclusions.

On page 24, ln 5 we will add this information: "The new estimates are 677 Gg $SO_2$ in the case of Kasatochi and 768 Gg $SO_2$ in the case of Sarychev that were injected into the altitude range 10–19 km."

- Page 24, ln 6: '... under OH background conditions'.

What do the authors refer to when saying 'OH background conditions'? That is not clear to me.

The OH levels have been derived in a CTM full chemistry run (2003–2006), without any feedback between the sulfur species and the OH concentrations.

For clarification the following will be added at page 24, ln 6: "... under OH background conditions (modelled OH climatology without any feedbacks between sulfur species and OH concentrations)."

[Figure]

[Figure]

**Fig. 1.**

---

## Author Response (AR2)

We thank the editor very much for the given recommendations. Please find our detailed responses to the comments by Referee #2 below.

*Comment by Referee #2:*

The authors have done a good job in correcting the mistakes and in answering my comments. However, there are a some mistakes (spelling and grammar) remaining and I would encourage the authors to carefully proof read the paper again. Also, maybe they want to give the paper to a native speaker (Terry Deshler who is a co-author on this paper) to find the remaining mistakes as this would improve the paper.

As requested for by Referee #2, we provide an updated version of the manuscript, in which we improved the English spelling and grammar to the best of our ability. We do not provide a list of changes, however, as they are marked by red/yellow (deleted/added) colours in the updated version of the manuscript.

One of the two concerns that remain is that they are not taking any measurement uncertainties into account when calculating mean values and especially the uncertainties on the mean values, which is not correct in my view. They did not convince me with their reply on my comment that this is not needed here or can't be done.

*Editor comment:*

I recommend that you either address that comment or explain in your reply why it currently appears current practice to not take into account measurement uncertainties.

We appreciate referee #2's comment on the calculation of uncertainties of the mean values and bias between in situ and MIPAS aerosol volume densities. However, we have intentionally not applied weighting of the data points by their inverse variances. The reason is this: the uncertainties depend on the atmospheric state. Larger aerosol loading increases the absolute errors of the balloon measurements, which are originally given as percentage errors. Furthermore, lower temperatures increase the noise in the MIPAS data. Lower temperatures often also go along with increased aerosol loading, as, compared to the gas-phase, the liquid-phase is favoured thermodynamically at lower temperatures. Weighting by inverse variances thus would give more weight to atmospheric conditions with low aerosol loading. The regression line thus would better represent conditions of low aerosol loading while large fit residuals would be accepted for higher aerosol loading. We want the regression line to be equally representative for high and low aerosol loading and thus do not weight the data points by their inverse variances.
We have expressed this in the text explicitly by introducing the sentence: "No weighting of the data points by their inverse error variances was applied in the calculation of mean in situ and MIPAS profiles. This method has been chosen in order to avoid representativeness problems, as the error variances correlate with the aerosol loading of the atmosphere and would thus cause a sampling artefact in the estimated bias and offset-correction."
In the updated version of Fig. 4b, the error estimates of the regression parameters and their covariance were used to estimate the resulting uncertainties of the altitude-dependent bias corrections. Additionally, we provide an updated version of the error bars included in Fig. 4a. These error bars are now considering the estimated measurement uncertainties, as correctly requested for by the referee. In the case of the in situ data, these are based on the given 40 % precision, while for the MIPAS profiles error bars are based on the noise error plus 10 % of the retrieved aerosol volume densities, as estimate of the remaining error contributions. The description in the text has been updated accordingly. The revised uncertainties of the mean profiles and the bias have not changed any of our conclusions or further results.

*Comment by Referee #2:*

My second concern, which became only apparent when looking at the revised paper, is that they seem to ignore temporal and spatial bias correction. They mention now that the differences

between model and measurements could be caused by biased means, however, they do not explain why they didn't do any spatial bias correction of the data before calculating the mean, although the state that the data are not uniformly distributed. I would encourage the authors to include one/two sentences on why they think that a spatial bias correction is not required and why they think that this wouldn't change the results much (I'm referring to page 26 from line 5 onward).

We agree with the referee that it would be ideal to have a dataset of uniformly distributed measurements also at lower altitudes in the tropics and directly after volcanic eruptions in which cases, e.g., the trace gas observations had to be filtered due to cloud/aerosol contamination of the infrared measurements (as already described in Höpfner et al., 2015). We are not aware of a reasonable method to correct the mean values calculated from the observations without application of external information. With respect to model inter-comparisons, there would be the possibility to sample the model only at the locations/times of the measurements. However, we refrain from doing so due to the following reasons: (1) there is still the possibility of not just sampling the measured structure exactly at the right position and time and (2) if one would perform such a detailed sampling, also the mean data derived from the model would be biased and, thus, not represent the true model mean. We decided to compare the measurement means to the model means but clearly indicated this in the updated text of the last submitted version of the manuscript (red text lines 13-23 on page 26 of the last submitted manuscript version with changes indicated). The main results of the paper are not affected by the effect of mentioned sampling artefacts, since those are based on times/parts of the comparison which are covered well by the observations or have explicitly been accounted for ($SO_2$ deficit in observations directly after the volcanic eruptions).

[revised manuscript text omitted]